# Spinal V2b neurons reveal a role for ipsilateral inhibition in speed control

Rebecca A Callahan[1], Richard Roberts[1], Mohini Sengupta[1], Yukiko Kimura[2], Shin-ichi Higashijima[2], Martha W Bagnall[1]*

[1]Department of Neuroscience, Washington University School of Medicine, St Louis, United States; [2]National Institute for Basic Biology, Okazaki, Japan

**Abstract** The spinal cord contains a diverse array of interneurons that govern motor output. Traditionally, models of spinal circuits have emphasized the role of inhibition in enforcing reciprocal alternation between left and right sides or flexors and extensors. However, recent work has shown that inhibition also increases coincident with excitation during contraction. Here, using larval zebrafish, we investigate the V2b (Gata3+) class of neurons, which contribute to flexor-extensor alternation but are otherwise poorly understood. Using newly generated transgenic lines we define two stable subclasses with distinct neurotransmitter and morphological properties. These V2b subclasses synapse directly onto motor neurons with differential targeting to speed-specific circuits. In vivo, optogenetic manipulation of V2b activity modulates locomotor frequency: suppressing V2b neurons elicits faster locomotion, whereas activating V2b neurons slows locomotion. We conclude that V2b neurons serve as a brake on axial motor circuits. Together, these results indicate a role for ipsilateral inhibition in speed control.

DOI: https://doi.org/10.7554/eLife.47837.001

*For correspondence: bagnall@wustl.edu

Competing interests: The authors declare that no competing interests exist.

## Introduction

Rhythmic, coordinated body movements require selective recruitment of motor neurons by spinal and supraspinal premotor circuits. Most vertebrates locomote via alternating left-right contractions that travel from rostral to caudal; tetrapods additionally alternate between flexors and extensors to regulate limb movements. Due in part to the technical challenges in identifying and manipulating specific classes of neurons in the spinal cord, the underlying circuitry of locomotion remains only poorly worked out.

Spinal neurons are broadly divided into ten superclasses arising from distinct progenitor domains of which six classes are considered to directly influence motor neurons (dI3, dI6, V0, V1, V2, V3) (*Arber, 2012*; *Bui et al., 2013*). Within these superclasses, cardinal neuron classes have been identified based on transcription factor expression and neurotransmitter identity (e.g., V2a/Chx10/excitatory; V2b/Gata3/inhibitory). Recently, it has become clear that many of these classes can be further subdivided into anywhere from 2 to 50 subclasses, based on anatomical and genetic distinctions, with as-yet unclear implications for circuit connectivity and function (*Song et al., 2018*; *Menelaou et al., 2019*; *Menelaou et al., 2014*; *Hayashi et al., 2018*; *Bikoff et al., 2016*).

Traditionally, patterned locomotion has been modeled as an alternation between excitation and inhibition, which dominate motor neurons during contraction and extension portions of the cycle, respectively (*Danner et al., 2017*). Recently, however, evidence from fish, frogs, and turtles has challenged the notion that inhibition is minimal during the contraction of the cycle, i.e., in-phase with excitation. Instead, inhibitory conductances appear to be significant both in- and out-of-phase (*Bagnall and McLean, 2014*; *Kishore et al., 2014*; *Berg et al., 2007*; *Petersen et al., 2014*; *Li and Moult, 2012*), suggesting that simultaneous recruitment of excitation and inhibition during the contraction is important for regulating motor neuron firing (*Petersen and Berg, 2016*).

In-phase inhibition is thought to derive from two spinal interneuron classes, the V1 and V2b populations. The V1 population includes Renshaw cells (*Alvarez et al., 2005*; *Sapir et al., 2004*), which provide recurrent inhibition onto motor neurons with potentially significant shunting effects (*Bhumbra et al., 2014*). To date, most analysis of drive from V2b neurons has focused on the shared contributions of V1s and V2bs to reciprocal inhibition governing flexor/extensor alternation in limbed animals (*Britz et al., 2015*; *Zhang et al., 2014*; *McCrea and Rybak, 2008*). However, this does not shed light on potential functions of ipsilateral inhibition in gain control for regulation of motor neuron firing *during* contraction, as opposed to suppression of motor neuron firing during extension.

In-phase inhibition increases in amplitude for faster locomotor movements (*Kishore et al., 2014*) suggesting a potential role in speed control. Here we investigated whether V2b neurons could indeed provide direct inhibition to motor neurons for speed control, taking advantage of the speed-dependent organization of zebrafish motor circuits (*McLean et al., 2007*; *Menelaou and McLean, 2012*; *McLean et al., 2008*). V2b neurons are good candidates for in-phase gain control because they are exclusively inhibitory in mouse and zebrafish (*Batista et al., 2008*) with ipsilateral, descending axons within the spinal cord (*Britz et al., 2015*; *Lundfald et al., 2007*). They arise from a final progenitor division that produces pairs of V2a and V2b neurons (*Kimura et al., 2008*). Given the role of V2a neurons in triggering motor output (*Kimura et al., 2013*; *Eklöf-Ljunggren et al., 2012*) particularly through speed-specific circuits for titrating levels of motor excitation (*Zhong et al., 2011*; *Crone et al., 2009*; *Ampatzis et al., 2014*; *Menelaou et al., 2014*), it seems plausible that their sister V2b neurons exert an opposing, inhibitory role in speed control. However, the V2b class has not been well characterized at anatomical or neurochemical levels outside of very early development.

Here, we define two subclasses of V2b neurons in larval zebrafish based on differential transmitter expression and anatomy and further show that these neurons directly inhibit axial motor neurons in speed-specific circuits. Optogenetic suppression of V2b activity elicits faster locomotion whereas optogenetic activation of V2b activity reduces tail frequency, consistent with a role for ipsilateral inhibition in speed control.

## Results

### Gata3 transgenic lines label V2b neurons

V2b neural identity is, in part, conferred by the developmental expression of the transcription factor Gata3 (*Karunaratne et al., 2002*; *Zhang et al., 2014*). To provide transgenic labeling of the V2b population, we generated two *gata3* transgenic lines, *Tg(gata3:loxP-DsRed-loxP:GFP)* and *Tg(gata3: Gal4)* from bacterial artificial chromosome (BAC) insertion transgenesis. These lines, along with *Tg (gad1b:GFP)* and *Tg(glyt2:loxP-mCherry-loxP:GFP)*, were shown to match endogenous gene expression with two-color in situ hybridization (see supplementary information, *Figure 1—figure supplement 1* and *Table 1*). Both lines label V2b neurons throughout the rostrocaudal extent of the larval zebrafish spinal cord (*Figure 1A*; *Tg(gata3:loxP-DsRed-loxP:GFP)* line shown). *Gata3*-driven fluorescent proteins are also broadly expressed in the brain, hindbrain, and assorted non-nervous system soft tissue including the pronephric duct (*Wingert et al., 2007*).

In a typical spinal segment, V2b soma position spanned the dorsoventral and mediolateral axes of the spinal cord (*Figure 1B*). Gata3 is expressed in not only V2b neurons but also the mechanosensory cerebrospinal fluid contacting neurons (CSF-cN) as well as in intraspinal serotonergic neurons

**Table 1.** Summary of in-situ hybridization transgenic line validation, including completeness and accuracy.

| Transgenic line | Completeness (%) | Sd | Accuracy (%) | Sd |
|---|---|---|---|---|
| *Tg(gata3:loxP-DsRed-loxP:GFP)* | 96.57 | 5.34 | 95.75 | 6.45 |
| *Tg(gata3:Gal4; UAS:GFP)* | 84.36 | 10.79 | 89.64 | 5.92 |
| *Tg(gad1b:GFP)* | 93.33 | 6.67 | 88.77 | 12.69 |
| *Tg(glyt2:loxP-mCherry-loxP:GFP)* | 86.51 | 7.26 | 92.21 | 2.70 |

DOI: https://doi.org/10.7554/eLife.47837.006

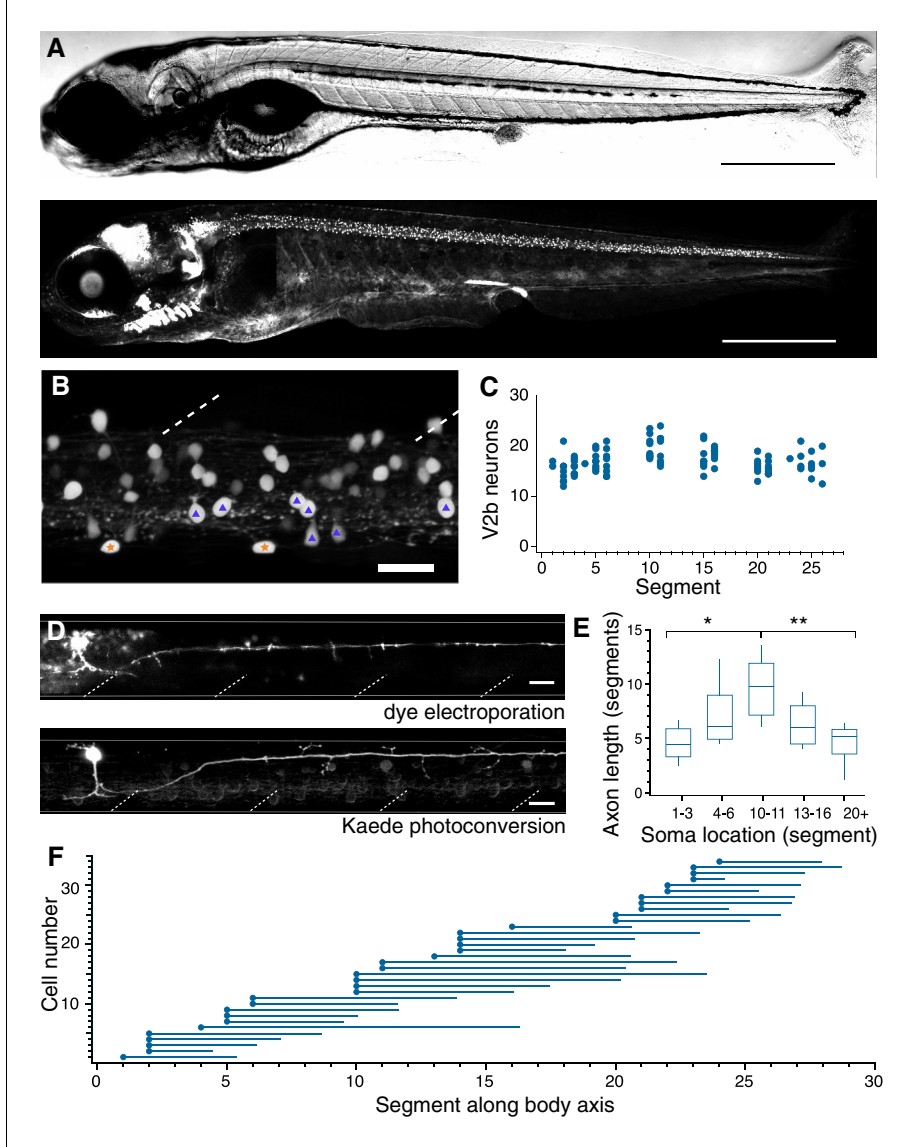

**Figure 1.** V2b neurons are found throughout the rostral-caudal axis of zebrafish spinal cord. (**A**) Transmitted DIC image (top) and confocal image (bottom) of a 5 dpf *Tg(gata3:loxP-dsRed-loxP:GFP)* animal. Scale bars = 0.5 mm. (**B**) Lateral view of a midbody spinal cord hemisegment; dashed lines mark muscle segments. In this and all subsequent figures, rostral is to the left and dorsal is to the top. Purple triangles mark CSF-cN neurons and orange stars mark ISN. Scale bar = 20 μm. (**C**) V2b cell counts per hemisegment quantified along the rostrocaudal body axis, n = 7 fish from three clutches. (**D**) Example cell morphology using two techniques to label single V2b axons: single-cell dye electroporation (top) and Kaede photoconversion (bottom). Scale bar = 20 μm. (**E**) Midbody V2b neurons extend axons through more segments than V2b neurons in other rostrocaudal locations. *p<0.01 (p=3.17×10⁻³); **p<0.001 (p=4.44×10⁻⁴). (**F**) Ball and stick plots indicate soma position and axon extension along the body axis for 35 V2b neurons in 12 animals from two clutches.

DOI: https://doi.org/10.7554/eLife.47837.002

The following source data and figure supplements are available for figure 1:

**Source data 1.** V2b axon length quantification.
DOI: https://doi.org/10.7554/eLife.47837.005

**Figure supplement 1.** Two-color fluorescent in-situ hybridization validates transgenic line expression patterns.
DOI: https://doi.org/10.7554/eLife.47837.003

**Figure supplement 2.** Example cell fills of the ventralmost cells in Tg(gata3:Gal4; UAS:Kaede) animals indicates intraspinal serotonergic neuron identity.
DOI: https://doi.org/10.7554/eLife.47837.004

(ISN, see Supplementary Information and *Figure 1—figure supplement 2*) (*Petracca et al., 2016*; *Montgomery et al., 2018*). Both classes were readily excluded from further anatomical analysis based on their distinct anatomical features: CSF-cNs exhibit large soma size, ventral position, and stereotyped extension into the central canal (triangles, *Figure 1B*), while ISNs have rectangular somata located along the ventral edge of the spinal cord (stars, *Figure 1B*). On average, each hemi-segment contained 17.2 + /- 2.5 (mean + /- SD) V2b neuron somata with relatively little variation from rostral to caudal segments (*Figure 1C*).

## V2b axons extend throughout the spinal cord

To visualize V2b axonal trajectories within the spinal cord, we labeled individual neurons via either single cell dye-electroporation or Kaede photoconversion in a *Tg(gata3:Gal4, UAS:Kaede)* line (*Figure 1D*) (*Ando et al., 2002*). No difference in axon length or trajectory was observed between the two methods. In all 59 neurons, the axon descended caudally and ipsilaterally, with an extent ranging from 2 to 15 segments. V2b axons originated on the ventral aspect of the soma and projected laterally into the white matter. Putative en passant boutons were seen as swellings distributed along the axon. Most V2b axons projected short collaterals into the soma-dense medial spinal cord along the axon extent. V2b dendrites extended from the main axon branch near the soma, (*Figure 1D*), similar to identified mixed processes in V2a neurons (*Menelaou et al., 2014*). However, in contrast to V2a neurons (*Menelaou et al., 2014*; *Azim et al., 2014*), no V2b neurons extended rostral axons beyond the segment of origin.

Single-cell Kaede photoconversions made at different positions along the rostrocaudal extent of the spinal cord revealed that axonal projections were longest for V2b somata located in the midbody range relative to V2b located in the rostralmost and caudalmost segments (*Figure 1E and F*, p=$3.17\times10^{-3}$ and p=$4.44\times10^{-4}$, ANOVA and Tukey's test). Overall, these data reveal that zebrafish V2b neurons exclusively innervate areas ipsilateral and caudal to the soma, with the greatest territory of axonal coverage originating from mid-body neurons with long axons.

## Neurotransmitter expression defines subpopulations of V2b neurons

Previous work has established that V2b neurons in embryonic zebrafish, as identified by Gata3 RNA expression, are exclusively inhibitory and predominantly GABAergic (*Batista et al., 2008*). To describe the neurotransmitter profile of V2b neurons in larvae, Gata3+ neurons were evaluated for coexpression with transgenic markers for *glyt2*, a glycine vesicular transport protein, and *gad1b*, a GABA synthesis enzyme. Nearly all larval V2b neurons expressed Glyt2 in 5 dpf larvae (*Figure 2A*), in contrast to embryonic stages. In contrast, Gad1b is expressed in approximately half of the V2b population (*Figure 2B*).

Inhibitory neurotransmitter switching is posited to occur at early developmental stages in zebrafish (*Higashijima et al., 2004a*). Therefore, we examined whether the variation of neurotransmitter expression in V2b neurons at 5 dpf represented a transient developmental stage or a stable pattern of expression. We assessed coexpression of the neurotransmitter markers in midbody V2b neurons at 5, 10, and 15 dpf (*Figure 2C*), after which V2b neurons are not reliably labeled by transgenic lines (data not shown). Gad1b and Glyt2 expression in V2b neurons remains unchanged across these ages, with ~52% of neurons expressing Gad1b and ~91% expressing Glyt2 (*Figure 2C*). These ratios indicate at least two subclasses of V2b neurons: those expressing Glyt2 but not Gad1b, and those expressing both Glyt2 and Gad1b. A possible third subclass of V2b neurons which express only Gad1b and not Glyt2 may exist; however, incomplete transgenic line label in *Tg(glyt2:GFP)* (see *Table 1*) could also account for Glyt2 expression below 100%.

Are GABAergic and non-GABAergic neurons distributed similarly throughout the neuraxis? By plotting dorsal-ventral (DV) position relative to spinal boundaries, we found that on average, GABAergic V2b somata are located slightly ventral to non-GABAergic V2b somata, but that both populations span the same DV range (*Figure 2D*). Therefore, soma position is not predictive of neurotransmitter expression. In the rostrocaudal axis, the percentage of GABAergic V2b cells is highest (~80%) in rostral segments, then decreases to ~50% by midbody and throughout the rest of the spinal cord. In contrast, Glyt2 robustly colabels with V2b cells throughout the entire spinal cord (*Figure 2E*). These data indicate that the Gad1b+ and Gad1b- populations comprise distinct and persistent subclasses. V2b neurons expressing both Glyt2 and Gad1b will be referred to as V2b-

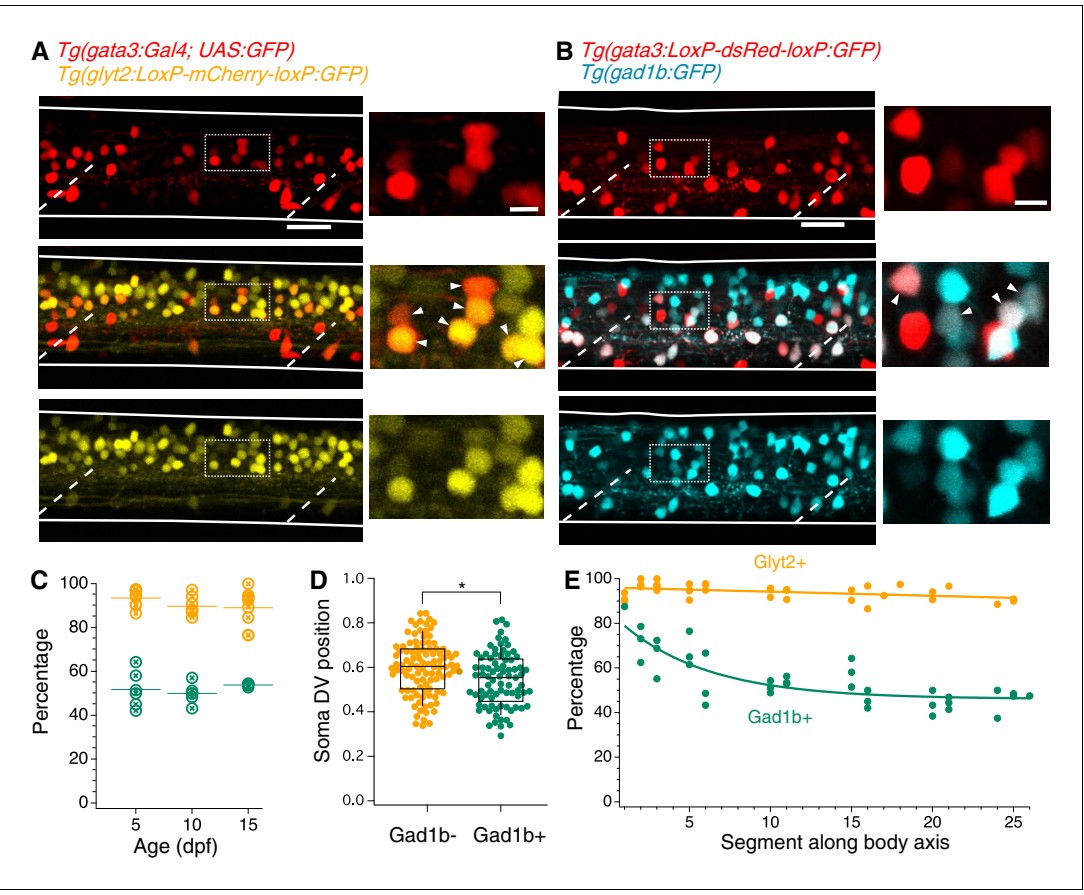

**Figure 2.** V2b neurons express the inhibitory neurotransmitter markers Glyt2 and Gad1b. (**A**) Lateral z-projection of a spinal cord hemisegment in a *Tg(gata3:Gal4,UAS:GFP;glyt2:loxP-mCherry-loxP:GFP)* (Gata3, top; Glyt2, bottom) double transgenic animal with composite image (middle). Dashed lines indicate muscle segments and solid lines indicate the spinal cord dorsal and ventral boundaries. Magnified inset, from dashed box, showing soma-level colocalization is shown to the right. Soma colocalization indicated with white arrowheads. Scale bar = 20 μm; inset 5 μm. (**B**) *Tg(gata3:loxP-DsRed-loxP:GFP;gad1b:GFP)* (Gata3, top; Gad1b, bottom) and dual-color composite image (middle). Magnified inset, from dashed box, is shown to the right. Soma colocalization indicated with white arrowheads. Scale bar = 20 μm; inset 5 μm. (**C**) Percentage of V2b neurons co-expressing Glyt2 (yellow) or Gad1b (green) is stable from ages 5–15 dpf, as measured in body segments 15–16. N = 6 animals at each time point, two clutches. (**D**) V2b soma position for Gad+ and Gad- neurons differs slightly in the dorsoventral axis, *p<0.01 (p=$1.87 \times 10^{-3}$), N = 108 neurons in N = 6 animals from two clutches, Student's t-test. (**E**) Percentage of V2b neurons co-expressing Glyt2 or Gad1b along the rostrocaudal body axis, N = 6 animals from two clutches.

DOI: https://doi.org/10.7554/eLife.47837.007

The following source data is available for figure 2:

**Source data 1.** Data underlying quantification of Gad1b/GlyT2 co-expression.
DOI: https://doi.org/10.7554/eLife.47837.008

mixed, in reference to their mixed neurotransmitter expression, whereas V2b neurons that solely express Glyt2 will be referred to as V2b-gly.

## Axonal morphology varies by subpopulation identity

The classic axonal morphology of zebrafish ventral longitudinal descending (VeLD) neurons is ventral, with little change in DV position from the onset (*Batista et al., 2008*). However, some V2b neuron fills exhibited axons with much more dorsal trajectories (e.g. *Figure 1D*). To resolve whether these represent different subclasses, we investigated axonal morphology of identified V2b-mixed and V2b-gly neurons using single-cell dye electroporation in the double transgenic line *Tg(gata3:loxP-*

*DsRed-loxP:GFP; gad1b:GFP)*, in which expression of GFP (Gad1b) differentiates between the mixed and glycinergic subclasses.

Although both V2b-gly and V2b-mixed neurons extend axons caudally and ipsilaterally, consistent with data in *Figure 1*, the DV position of their axons was different. GABAergic V2b-mixed neurons projected axons ventrally along the spinal cord, with an average axon location found between 0.24–0.33 DV (example, *Figure 3A*). In contrast, axons from V2b-gly neurons typically make an initial ventral dip but then turn more dorsally, ranging from 0.31 to 0.65 in the DV axis (example, *Figure 3B*). Traces from all filled neurons are shown in *Figure 3C*, and averaged trajectories in *Figure 3D*. Somata were filled in segments ranging from 14 to 18; the traces are shown aligned at the soma for ease of visualization.

Other features of anatomy also varied between the two subtypes. The axon DV position of V2b-gly neurons is positively correlated to the soma DV position, that is a more dorsal soma projects a more dorsally positioned axon (*Figure 3G*). However, this trend is not realized for V2b-mixed cells, which project axons ventrally to a narrow spinal cord region regardless of soma position. Putative en passant boutons were found in both cell populations. Most filled axons (22/24) extended vertical collaterals from the main axon. The number of collaterals per axon did not significantly vary between populations (V2b-mixed, median = 3, range 0–5; V2b-gly, median = 5, range 0–23, Mann-Whitney Wilcoxon test p=0.056). However, collaterals of V2b-mixed and V2b-gly axons cover largely distinct DV regions of the spinal cord (*Figure 3H*).

What is the significance of differential DV axon trajectories between V2b-gly and V2b-mixed subclasses? Previous work has shown that motor neurons active during fast movements are located more dorsally within the spinal cord, whereas those for slower movements are located more ventrally (*McLean et al., 2007*). Therefore, we compared population averages of the V2b-gly and V2b-mixed axons (*Figure 3D*) to a plot of motor neuron dendritic territory (*Figure 3E*; see Materials and methods). Notably, V2b axon position of the two classes overlaps with two peaks in the motor neuron density profile. Consequently, we next investigated the direct influence of V2b neurons on motor neurons.

## V2b subpopulations provide differential inputs to fast and slow circuits

Anatomical evidence indicates that V2b neurons make contact onto limb motor neurons where they are partially responsible for enforcing flexor/extensor alternation (*Britz et al., 2015*; *Zhang et al., 2014*). However, to date there are no physiological recordings of synaptic connections from V2bs onto motor neurons or other targets. We first validated that optogenetic stimulation in the *Tg (gata3:Gal4; UAS:CatCh)* line was sufficient to elicit action potentials in V2b neurons (*Figure 4A*). We then targeted spinal motor neurons for in vivo recording in *Tg(gata3:Gal4, UAS:CatCh)* larvae at 4–6 dpf (*Figure 4B*). Optogenetic activation of V2b neurons with a 20–50 ms pulse of light delivered 3–7 segments rostral to the recording site elicited robust IPSCs in motor neurons (*Figure 4D and E*). Synaptic conductance amplitudes exhibited a median of 139 pS (25–75% range, 97–174 pS). Although the *Tg(gata3:Gal4)* line labels CSF-cNs in addition to V2bs (*Figure 1B*), CSF-cNs exhibit short ascending axons that do not contact motor neurons other than the CaP (*Hubbard et al., 2016*). To validate that V2b neurons are providing these inhibitory inputs, we used a digital micromirror device to deliver targeted squares of light stimuli (~20 µm x 20 µm) to dorsal spinal cord areas containing V2b but not CSF-cN or ISN somata. These localized stimuli still elicited reliable IPSCs in both primary and secondary motor neurons (*Figure 4—figure supplement 1A*; *Figure 4—figure supplement 1B*; *Figure 4—figure supplement 1C*). As a second control, recordings were also made in a subset of animals with strong CatCh expression in CSF-cNs and negligible expression in V2b neurons (see Materials and methods and *Figure 4—figure supplement 1D*). In these recordings, even full-field light stimulation did not evoke IPSCs in motor neurons (*Figure 4—figure supplement 1E*). Together these results indicate that the optogenetically elicited inhibitory inputs arise from monosynaptic V2b to motor neuron connections.

The striking difference in dorsal-ventral targeting of V2b-gly and V2b-mixed axonal trajectories (*Figure 3D*) suggests a potential relationship with the well-described dorsal-ventral distribution of motor neurons based on size and speed at recruitment. Large motor neurons with low input resistance are located dorsally within the motor pool and are recruited for the fastest speeds of swimming, whereas more ventrally located motor neuron somata exhibit higher input resistance and are recruited during slower movements (*McLean et al., 2007*; *Menelaou and McLean, 2012*).

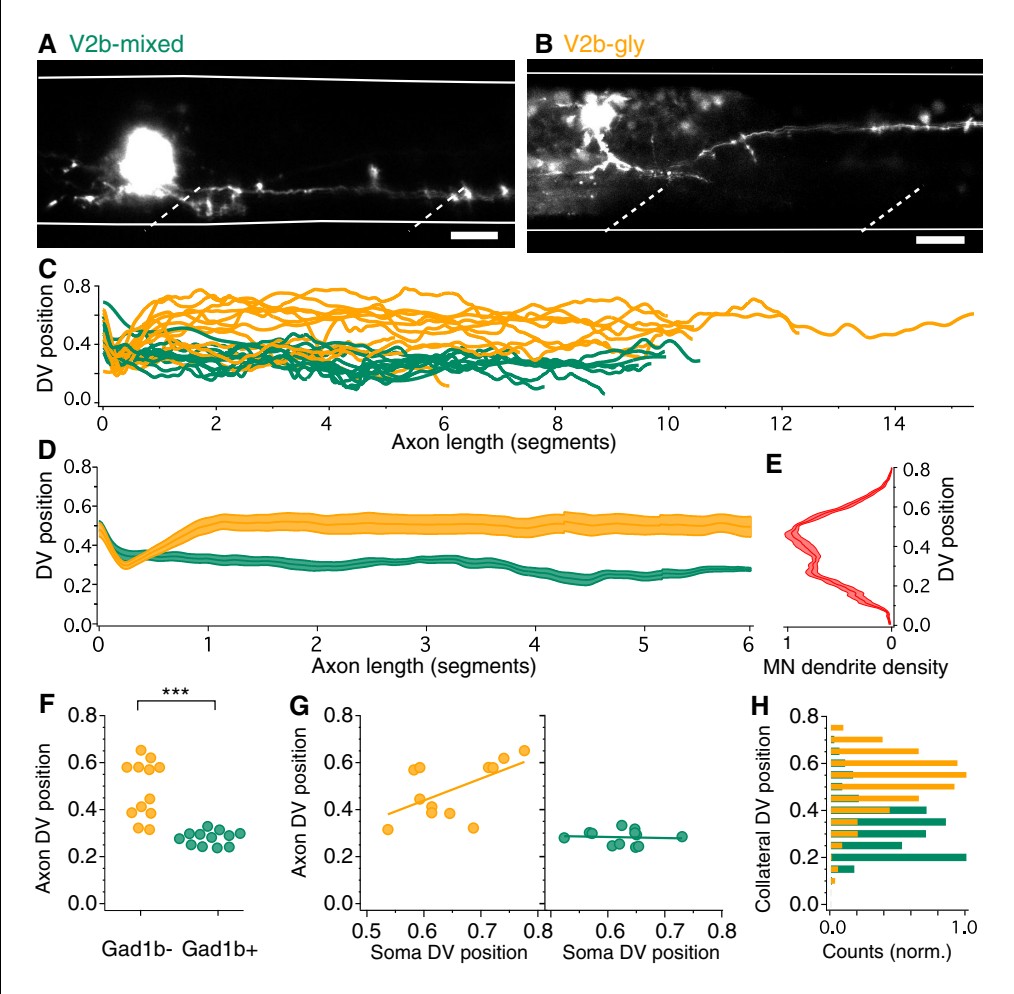

**Figure 3.** V2b-gly and V2b-mixed neurons have distinct axon morphology and innervation territories. (**A**) Examples of a V2b-mixed (*Tg(gad1b:GFP)*+) and a (**B**) V2b-gly (*Tg(gad1b:GFP)*-) single-cell dye fill. Dashed lines indicate muscle segments and solid lines indicate the spinal cord dorsal and ventral boundaries. Scale bars = 20 μm. (**C**) Axon traces for V2b neurons, aligned at the segment of origin, relative to the spinal cord dorsoventral boundaries (V2b-mixed, green, N = 12; V2b-gly, orange, N = 12 from eight clutches). All axons were exclusively descending. (**D**) Mean and SEM of V2b-gly and V2b-mixed axon trajectories. (**E**) Motor neuron dendrite fluorescence intensity, measured in *Tg(mnx:GFP)*, relative to the same dorsoventral landmarks. (**F**) Mean axon position for each traced axon. ***p<0.0001 (p=8.23×10$^{-5}$), Student's t-test. (**G**) Average axon position of V2b-mixed (green, left) and V2b-gly (orange, right) relative to soma position for each neuron. A correlation between soma position and axon position is observed for V2b-gly but not V2b-mixed neurons. V2b-gly: $r^2$ = 0.33, p<0.05, V2b-mixed: $r^2$ = 0.0059, p=n .s. (**H**) Axon collaterals of V2b-gly neurons also innervate more dorsal spinal cord territory than V2b-mixed axons.

DOI: https://doi.org/10.7554/eLife.47837.009

The following source data is available for figure 3:

**Source data 1.** Data underlying quantification of axon position.
DOI: https://doi.org/10.7554/eLife.47837.010

Accordingly, we tested whether the glycinergic and GABAergic components of the IPSC differed between fast and slow motor neurons. Bath application of 10 μM strychnine to block glycine receptors abolished a median of 91% of the V2b-evoked IPSC in fast motor neurons, but only 71% of the V2b-evoked IPSC in slow motor neurons (*Figure 4D and E,C*; p=0.003, Wilcoxon Rank test). Similar results were obtained with measurement of the charge integral (over 100 ms: 91% blocked by strychnine in fast motor neurons; 66% block in slow motor neurons). The GABA$_A$ receptor antagonist

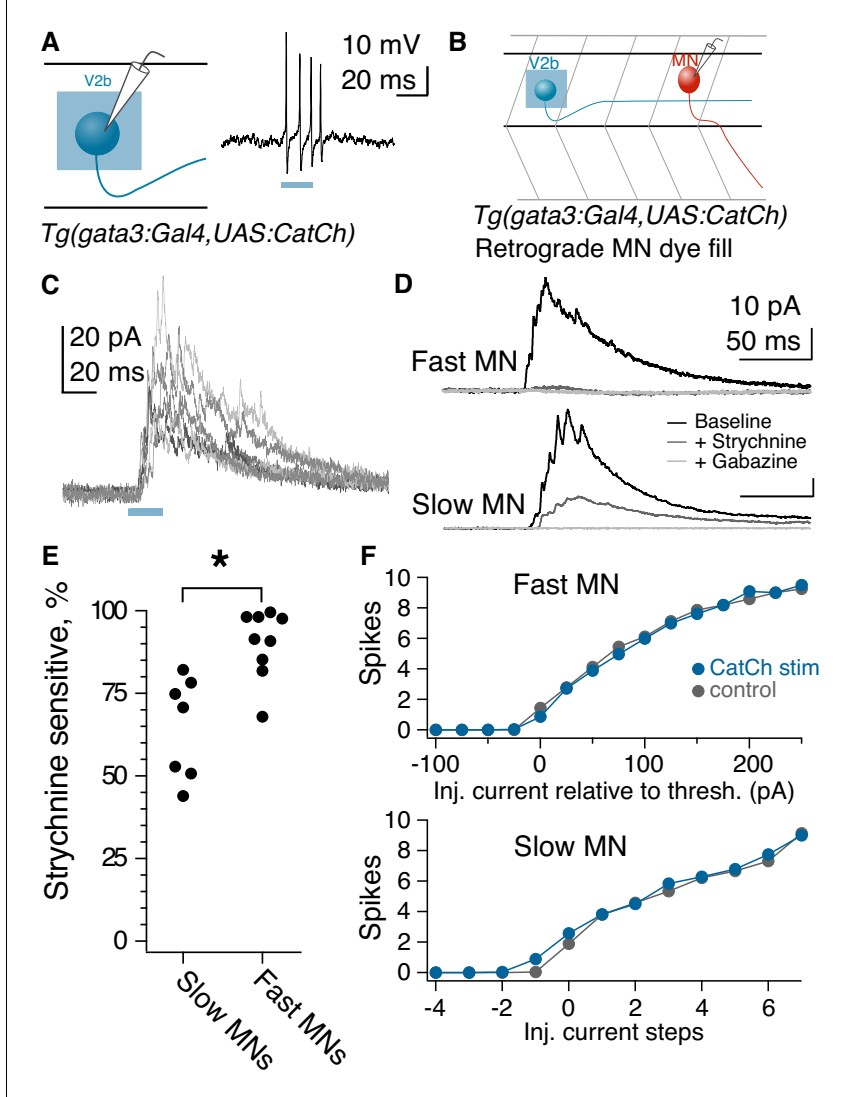

**Figure 4.** Fast motor neurons receive predominantly glycinergic V2b inputs, whereas V2b synaptic inputs to slow motor neurons are mediated by both GABA and glycine receptors. (A) Schematic of recording to validate CatCh expression in V2b neurons and cell-attached recording from a V2b neuron expressing CatCh during a 20 ms illumination epoch. Note that evoked action potentials outlast the duration of illumination, presumably due to membrane depolarization and/or Ca influx. (B) Schematic illustrating whole-cell recordings from motor neurons paired with optogenetic stimulation of V2b neurons. (C) Six overlaid sweeps showing ISPCs barrages recorded in a motor neuron in response to optogenetic activation of V2b neurons. Blue bar represents the light stimulus. All recordings were carried out in the presence of NBQX. (D) Average IPSC responses to light stimulation in fast (top) and slow (bottom) motor neurons, as identified by soma location and input resistance. Response during baseline (black), after application of strychnine (dark gray), and after additional application of gabazine (light gray). In all cases, the IPSC was entirely abolished by the combination of strychnine and gabazine. N = 7 slow motor neurons from two clutches, N = 9 fast motor neurons from four clutches. (E) Percentage peak current reduction by strychnine in fast and slow motor neurons. *p<0.01 (p=2.0×10$^{-3}$). (F) Elicited spike counts in motor neurons (fast motor neurons, top; slow, bottom) during 30 ms depolarizing steps, ranging from sub- to supra-threshold, with (blue) and without (black) concurrent optogenetic activation of V2bs. Input resistance varied across slow motor neurons, and accordingly the number of current steps relative to threshold is shown rather than the actual injected current (stepsize range: 5–25 pA).

DOI: https://doi.org/10.7554/eLife.47837.011

The following source data and figure supplement are available for figure 4:

**Source data 1.** Characteristics of V2b-evoked inhibition in motor neurons, source data.

DOI: https://doi.org/10.7554/eLife.47837.013

*Figure 4 continued*

**Figure supplement 1.** Optogenetically evoked IPSCs originate from V2b neurons, not CSF-cNs.
DOI: https://doi.org/10.7554/eLife.47837.012

gabazine (SR-95531, 10 µM) eliminated the remaining IPSC in all cases. Therefore V2b-mediated inhibition onto fast motor neurons is carried out predominantly by glycinergic synapses, whereas V2b inhibition onto slow motor neurons is carried by mixed glycinergic/GABAergic transmission. These results are consistent with the idea that V2b-gly preferentially inhibit more dorsally located fast motor neurons, whereas V2b-mixed inhibit the more ventrally located slow motor neurons.

Does V2b-mediated inhibition suppress spiking in motor neurons? We tested this by delivering 30 ms pulses of depolarizing current injection to motor neurons, with or without conjunctive optogenetic activation of V2b neurons in the *Tg(gata3:Gal4; UAS:CatCh)* line. Interestingly, V2b activation did not change thresholds or gain relationships on average, in both fast and slow motor neurons (*Figure 4F*). Within fast motor neurons, 3/14 neurons did exhibit a clear increased threshold to spike during V2b activation, and in two of those neurons, strychnine completely abrogated the effects (the third was not tested). However, the overall results indicate that V2b suppression of motor neuron spiking is not robust, despite the clear evidence of connectivity. These findings may be attributable to the putative location of V2b synapses on motor neuron dendritic arbors and consequent effects on dendritic integration rather than somatically-elicited spiking.

Some spinal premotor neurons form synaptic connections within their own populations, suggestive of speed- or state-related 'gears' (*Chopek et al., 2018*). Optogenetic activation of rostrally-located V2b neurons evoked IPSCs in 11/21 (52%) mid-body V2b neurons (*Figure 5A and B*). The median conductance of individual IPSCs was 158 pS (25–75% range, 84–203 pS). Application of strychnine blocked a median of 80% of the V2b-evoked IPSC, while the remainder was abolished by gabazine (*Figure 5B and C*). Thus, some V2b neurons inhibit other members of the V2b pool, forming a disinhibitory pathway.

## V2b cell physiology does not distinguish between subtypes

Intrinsic physiological characteristics, including input resistance and spiking properties, can be used to subdivide some spinal interneuron populations into distinct subpopulations (*Borowska et al., 2013*; *Song et al., 2018*; *Bikoff et al., 2016*). We examined whether the V2b-gly and V2b-mixed subgroups exhibited differences in intrinsic physiology by targeting whole-cell recordings to these neurons. V2b neurons were silent at rest, in contrast to CSF-cNs which exhibited spontaneous spiking (data not shown). Spikes were elicited by depolarizing current steps, (*Figure 5D and E*) which usually led to one or a few spikes, with only 3/10 V2b-gly and 3/9 V2b-mixed neurons able to sustain spiking across the step. Spikes were typically small in amplitude, similar to what is seen in zebrafish V1 neurons (*Higashijima et al., 2004b*), likely reflecting action potential initiation at some electrotonic distance from the soma. There was no difference in input resistance (*Figure 5F*) or spike shape (*Figure 5G*) between the V2b-gly and V2b-mixed neurons. Therefore, the two V2b subpopulations are indistinguishable at the level of intrinsic physiology despite their anatomical differences.

## Optogenetic V2b suppression increases tail beat frequency

What are the functional consequences of V2b inhibition onto ipsilateral motor circuits? To better understand this role, we carried out high-speed behavioral recordings during optogenetic inactivation of V2b neurons with a light-gated Cl⁻ channel, ZipACR (*Bergs et al., 2018*). To eliminate contributions from Gata3 expressing neurons in the brain, we used a spinally transected preparation. Tail movements were induced pharmacologically with application of N-methyl-d-aspartate (NMDA, 200 µM) (*McDearmid and Drapeau, 2006*). NMDA induces tail movements with episodic, left-right alternations that mimic the natural beat-and-glide swims of 5 dpf larvae (*McDearmid and Drapeau, 2006*; *Wiggin et al., 2014*; *Wiggin et al., 2012*).

Spinal V2b neurons, CSF-cNs and ISNs are labeled in BAC-generated Gata3 transgenic lines (*Figure 1B*). However, a CRISPR-generated *Tg(gata3:ZipACR-YFP)* knock-in shows robust expression of the fluorescent ZipACR protein in V2b neurons and ISN but only sparse, dim expression in CSF-

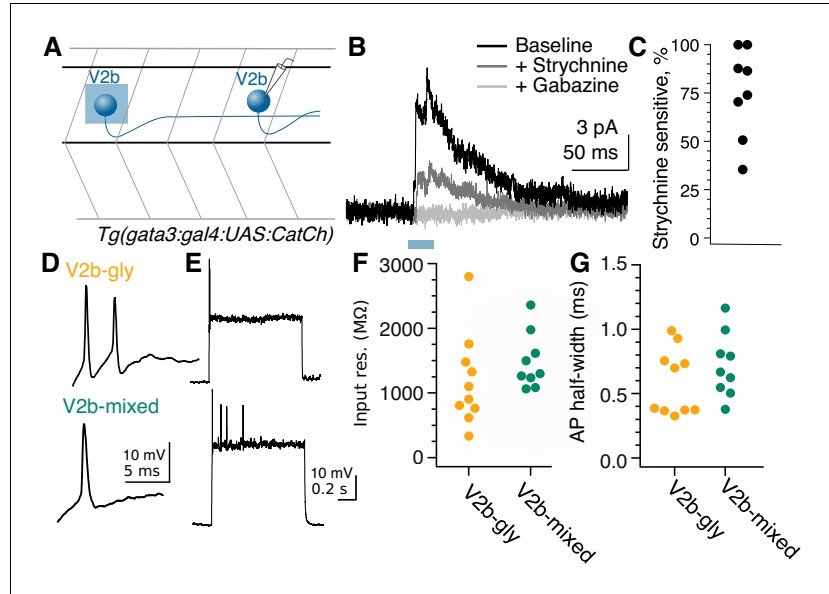

**Figure 5.** Rostral V2b neurons inhibit more caudal V2b neurons, providing circuit disinhibition; V2b-gly and V2b mixed populations are physiologically indistinguishable. (**A**) Experimental schematic for V2b-to-V2b connectivity recordings. (**B**) Evoked IPSCs recorded in an example V2b neuron in response to optogenetic stimulation of more rostral V2b neurons, black, and the response after the successive addition of glycine and GABA$_A$ receptor antagonists, dark gray and light gray traces respectively. The blue bar represents the duration of optogenetic stimulation. (**C**) Percentage peak current sensitivity to strychnine. (**D**) Example action potential magnified from (**E**) responses to step depolarizations in both classes of V2b neurons. Most recorded neurons in both groups could not sustain action potentials across a step. (**F**) Input resistance measured via hyperpolarizing test pulse. N = 10 Gad- (orange), 9 Gad+ (green), from 5 and 6 clutches, respectively. Two neurons were excluded that were spontaneously firing on patch, both of which were round and near the canal, as putative CSF-cns. (**G**) Action potential half-widths are not significantly different between the two groups.

DOI: https://doi.org/10.7554/eLife.47837.014

The following source data is available for figure 5:

**Source data 1.** Characteristics of V2b neuron physiology.

DOI: https://doi.org/10.7554/eLife.47837.015

cNs (*Figure 6A*). CSF-cNs expression was observed exclusively in the apical extension into the central canal but not the soma (*Figure 6A and Figure 6—figure supplement 1*).

We first validated the efficacy of the ZipACR construct in suppressing V2b firing under high-intensity light (*Figure 6—figure supplement 2A and B*, n = 4). Under lower-intensity light conditions, identical to those of the behavioral recordings, action potentials were completely suppressed in 4 out of 6 V2b cells and partially suppressed in one additional cell (*Figure 6B and C*), in *Tg(gata3: ZipACR-YFP; gata3:loxP-DsRed-loxP:GFP)* animals. In contrast, identical stimulation partially suppressed spiking in only 1 of 5 CSF-cNs (*Figure 6C* and *Figure 6—figure supplement 2C*). Therefore, we used these light stimulation parameters, under which V2b neurons are mostly if not entirely suppressed whereas CSF-cNs are not substantially affected, to carry out behavioral experiments assessing the effects of suppressing V2b neurons on locomotion.

Animals were head-embedded with a free tail and high-speed (200 Hz) recordings were acquired to capture fictive locomotion. Swim dynamics were recorded and evaluated both with and without optogenetic stimulation (*Figure 6D*). Kinematic analysis of the high-speed video was performed with code adapted from *Severi et al. (2018)*. The total tail displacement and quantity of tail movements did not significantly differ during optogenetic stimulation (*Figure 6D and F*). Strikingly, tail beat frequency (TBF), a proxy for locomotor speed (*Müller and van Leeuwen, 2004*), increased in *Tg(gata3: ZipACR-YFP)+* animals during light stimulation but not in their ZipACR negative clutchmates (*Figure 6E and G*, paired Student's t-test p=0.0025). The average TBF change was 1.4 Hz (control TBF mean = 13.4 Hz, stimulation TBF mean = 14.7 Hz) and was robustly observed in animals from

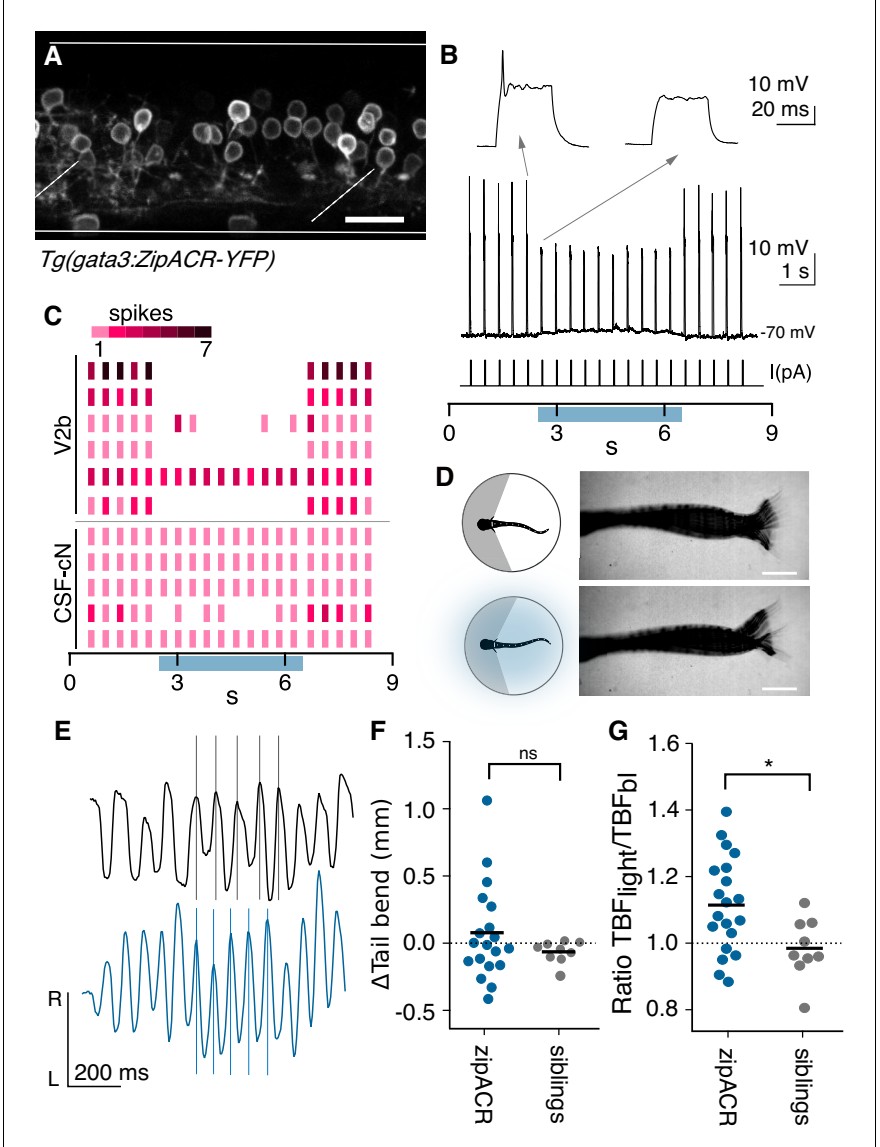

**Figure 6.** Optogenetic suppression of V2b activity leads to increased locomotor speeds. (**A**) Z-projection of *Tg (gata3:zipACR-YFP)* over one full segment of spinal cord showing expression in V2b but not CSF-cN somata. CSF-cN apical extentions show some YFP expression. See also *Figure 6—figure supplement 1*. Scale bar = 20 μm. (**B**) A whole cell recording during repeated current steps (30 ms duration) is shown for an example V2b neuron in a *Tg (gata3:zipACR-YFP)* animal. Blue bar indicates period of optical stimulation. An expanded view of current steps before and during optical stimulation are shown above with arrows. Recordings indicate that current steps normally elicit bursts of action potentials, but coincident optogenetic suppression prevents spiking, yielding only subthreshold depolarizations. (**C**) Raster plot of action potentials for 6 V2b cells and 5 CSF-cN cells summarizes optogenetic suppression across cell types. Color value represents number of spikes elicited during each current step. 5/6 V2b neurons were mostly or entirely suppressed, whereas only 1/5 CSF-cN were affected. (**D**) Schematic of behavioral recording depicting the NMDA-induced tail movements of spinalized head-embedded animals without and with optogenetic stimulation. Image overlay of 100 ms of tail movements without and with light stimulation in a *Tg(gata3:zipACR-YFP)* animal show similar amplitude tail displacement during swim. Scale bar = 0.5 mm. (**E**) Tracked left-right tail position during recordings with (blue) and without (black) optical stimulation for the same *Tg(gata3:zipACR-YFP)* animal. Lines for each recording are aligned to consecutive peaks in the baseline trace to illustrate the phase advance and increased tail beat frequency during optogenetic stimulation. (**F**) Average change in tail bend amplitude between stimulation and control recordings during swim movements for each animal, ns (p=0.14). (**G**) Ratio of average TBF during stimulation to baseline TBF for each

*Figure 6 continued*
animal, cohort averages shown with black dash. N = 20 *Tg(gata3:zipACR-YFP)* and N = 9 siblings. *p<0.01
(p=7.23×10⁻³).
DOI: https://doi.org/10.7554/eLife.47837.016
The following source data and figure supplements are available for figure 6:
**Source data 1.** Quantification of zipACR suppression.
DOI: https://doi.org/10.7554/eLife.47837.019
**Figure supplement 1.** Anatomy of *Tg(gata3:zipACR-YFP)* expression.
DOI: https://doi.org/10.7554/eLife.47837.017
**Figure supplement 2.** Additional recordings in *Tg(gata3:zipACR-YFP)* animals.
DOI: https://doi.org/10.7554/eLife.47837.018

three clutches. In contrast, elimination or genetic silencing of CSF-cN reduces TBF during acoustically-evoked escapes (*Böhm et al., 2016*). Furthermore, in larval zebrafish neither serotonin nor serotonergic receptor antagonists were found to influence locomotor rhythms or tail beat frequency (J. Montgomery and M. Masino, personal communication, July 2019), but instead influenced periods of activity (*Montgomery et al., 2018*; *Brustein et al., 2003*). Thus stimulation of ISNs is unlikely to contribute to the observed TBF changes. We conclude that suppression of V2b neurons increases locomotor frequency.

## V2b activation reduces TBF during evoked swims

We next investigated how the converse manipulation, V2b activation, influences locomotor rhythms. *Tg(gata3:Gal4; UAS:CatCh)* animals exhibited variable non-specific expression of CatCh in some muscle fibers, precluding experiments on freely swimming or head-embedded preparations. Instead, we recorded fictive swim from the motor ventral root in paralyzed, intact preparations. Brief electrical stimuli were used to elicit fictive locomotion, and in alternate trials were paired with localized optogenetic stimulation at 3–4 segments rostral to the recording (*Figure 7A*). Optogenetic stimulation rostral to the ventral root recordings, targeting V2b neurons that provide inhibition to the recorded segment, led to a decrease in the first inter-burst interval (IBI) relative to trials without light stimulation (*Figure 7D and E*, paired Student's t-test p=1.72×10⁻⁴). IBIs across the entire swim bout, which progresses from faster to slower, were also reduced in frequency (*Figure 7F*, p=1.51×10⁻³, paired Student's t-test) though to a lesser degree than the first IBI (Mann-Whitney Wilcoxon test p=1.23×10⁻³).

Several control experiments were used to ensure that these effects were attributable to V2b activation, rather than CSF-cN activation. First, to test whether connections from CSF-cNs could slow locomotion, optogenetic activation was delivered caudally to the recorded segment, to target the predominantly ascending axons of CSF-cNs. Caudal stimulation did not affect IBI frequency (*Figure 7G and H*, paired Student's t-test, p=0.495). Additionally, we repeated these experiments in a set of *Tg(gata3:Gal4; UAS:CatCh)* animals with low CatCh expression in V2b neurons to account for CSF-cN contributions (<3 V2b neurons per hemisegment, akin to that used in *Figure 4—figure supplement 1*) (*Figure 7C*). Optogenetic stimulation in low V2b CatCh expressing animals did not lead to a significant change in tail frequency (*Figure 7E*, paired Student's t-test p=0.174, CatCh- sibling control group p=0.427). Finally, to determine whether visually-induced effects contributed to the locomotor slowing, we delivered a red-shifted excitation (~550 nm) incapable of activating CatCh to rostral segments in the normally-expressing *Tg(gata3:Gal4; UAS:CatCh)* line. Tail beat frequencies of the red-shifted stimulation did not vary from controls (*Figure 7G and H*, paired Student's t-test, p=0.467). Activation of V2b neurons in the original experiment did reduce swim bout duration (*Figure 7I*, Student's t-test p=4.65×10⁻³). However, given the expression of CatCh in ISNs and their known role in influencing swim bout incidence (*Montgomery et al., 2018*) it is not possible to ascertain whether this result arises from V2b inhibition, ISN serotonergic modulation, or a superposition of both. Bout length was not significantly altered in animals with low V2b but strong CSF-cN CatCh expression (*Figure 7I*, Student's t-test p=0.113), though we note that CSF-cN activation after swim onset (as opposed to throughout) has been shown to reduce bout duration (*Fidelin et al., 2015*).

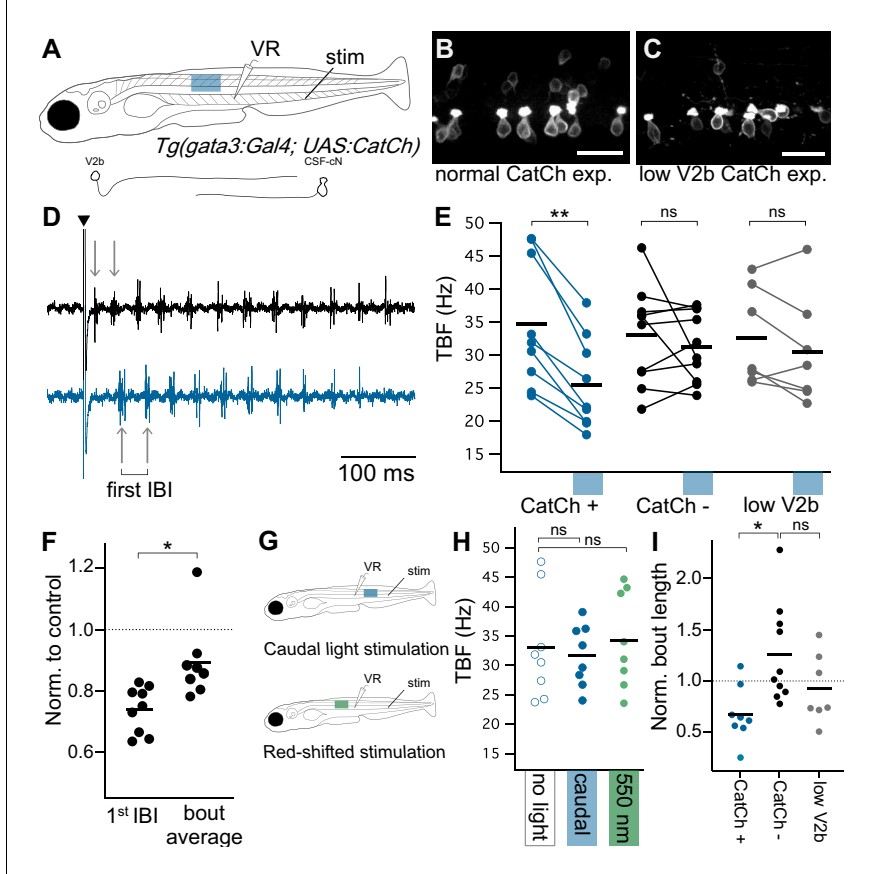

**Figure 7.** Optogenetic activation of Gata3+ neurons leads to decreased locomotor speeds. (A) Schematic of ventral root recording and electrical tail stimulation with rostrally-positioned optogenetic stimulation in *Tg(gata3: Gal4; UAS:CatCh)* animals. Representation of V2b and CSF-cN cellular morphology. Confocal z-projections show CatCh expression in one segment of spinal cord in (B) an animal with broad CatCh expression in V2b and CSF-cN cell types and (C) an animal with low CatCh expression in V2b neurons. Scale bars = 20 μm. (D) Example recording of ventral root motor activity after electrical stimulation (caret) showing the swim rhythm in the control (black) and optogenetically stimulated trial (blue) for the same animal. Arrows distinguish the first inter-burst interval (IBI), showing a difference in tail beat frequency. (E) Multi-trial average of first IBI compared between trials with (blue x-axis bar) and without optogenetic stimulation for animals with broad CatCh expression (CatCh+), sibling controls (CatCh-), and low V2b-expressing CatCh+ animals (low V2b). Dashes represent cohort averages. **p<0.001 (p=1.72×10$^{-4}$). (F) TBF during rostral optogenetic stimulation normalized to no-light control for the first IBI and swim bout average IBI. *p<0.01 (p=1.23×10$^{-3}$). (G) Experimental schematic of additional control experiments including a caudally-located stimulation (top) and a red-shifted light stimulation (bottom). (H) Multi-trial average of first IBI in *Tg(gata3:Gal4; UAS:CatCh)* animals with broad CatCh expression show that caudal stimulation (blue) and red-shifted light stimulation controls (green) do not vary from baseline non-optogenetic stimulation IBIs (open circles). (I) Bout duration during rostral optogenetic stimulation is shown normalized to the baseline bout duration *p<01 (p=4.65×10$^{-3}$). Broad CatCh-expressing animals, n = 9, CatCh- siblings, n = 10, low V2b expression, n = 7.
DOI: https://doi.org/10.7554/eLife.47837.020

The following source data is available for figure 7:

**Source data 1.** Quantification of behavioral effects of V2b activation.
DOI: https://doi.org/10.7554/eLife.47837.021

All together these experiments demonstrate that stimulating V2b neurons reduces the speed of rhythmic tail movements during swim. In combination with our finding that V2b suppression increased swim frequency, these results provide evidence that V2b neurons play a role in speed control in motor circuits by acting as a brake on locomotor frequency.

## Discussion

In this study we demonstrate that V2b neurons exert direct control over axial musculature in the larval zebrafish. The V2b population comprises two stable subclasses, defined by neurotransmitter identity: one subclass is exclusively glycinergic and the other mixed glycinergic/GABAergic. These distinct V2b-gly and V2b-mixed subclasses preferentially inhibit fast and slow motor neurons, respectively, analogous to the speed-dependent connectivity found in a diverse range of zebrafish spinal interneurons (*McLean and Fetcho, 2009*; *McLean et al., 2008*; *Ampatzis et al., 2014*; *Svara et al., 2018*). Moreover, we found that V2b activity serves as a brake on locomotor speeds: driving V2b neurons slows locomotion whereas suppressing V2b neurons speeds it up. Together, these results indicate that inhibition from V2b neurons is not restricted to enforcing agonist-antagonist muscle coordination but also influences locomotor speed through in-phase modulation of axial motor circuits.

### V2b conservation across species

We demonstrated that V2b neurons are inhibitory and extend axons ipsilaterally and caudally in zebrafish (*Figures 1* and *3*) similar to V2b neurons in mice (*Lundfald et al., 2007*; *Britz et al., 2015*). Gata3+ V2b neurons are widely present in vertebrates but have also been identified in the nerve cord of a marine annelid, indicating an ancestral persistence in motor circuitry (*Karunaratne et al., 2002*; *Vergara et al., 2017*). V2b neurotransmitter profiles appear to vary throughout development and across species. Gata3-expressing cells in the embryonic 24 hr post fertilization zebrafish are predominantly GABAergic with a smaller subset expressing or co-expressing glycine (*Batista et al., 2008*), an inversion of our finding that V2b cells in 5–15 dpf zebrafish are all glycinergic with approximately half co-expressing GABA. Early in development, murine V2b neurons broadly co-express GABA and glycine (*Zhang et al., 2014*). By P0 in mouse, however, nearly all V2bs are glycinergic and ~25% are GABAergic (*Lundfald et al., 2007*), which is broadly similar to our results. In zebrafish, these two subclasses persist out to 15 dpf, implying that they are stable identities.

Consistent with the idea that V2b-gly and V2b-mixed are distinct identities, the two subclasses exhibit different axon trajectories in the DV axis, perhaps indicating responsiveness to different axon guidance cues. In mouse, V2b subpopulations have not been directly shown. However, the differential expression of the transcription factors Gata2/Gata3/BhlhB5 in non-overlapping neural subsets may imply their presence (*Francius et al., 2013*). More broadly, our finding of subclasses in the V2b population is parallel to previously identified genetically and anatomically distinct subclasses within the V0, V1, and V2a populations in the mouse and zebrafish (*Bikoff et al., 2016*; *Hayashi et al., 2018*; *Menelaou et al., 2014*; *Song et al., 2018*; *Björnfors and El Manira, 2016*).

### Do V2b-gly and V2b-mixed populations match existing zebrafish neural classes?

Historically, zebrafish spinal neurons have been classified by anatomy. V2b (Gata3+) neurons are thought to correlate to ventral longitudinal (VeLD) neurons, an anatomically defined cell class with a characteristic longitudinal ventral-positioned axon (*Batista et al., 2008*; *Hale et al., 2001*; *Bernhardt et al., 1992*). How do the subpopulations we have described here relate to the VeLD population? Based on the ventral axon morphology and GABA co-expression, the V2b-mixed subtype represents a matured version of the embryonic VeLD neurons. In contrast, V2b-gly neurons are distinct in morphology and neurotransmitter profile from VeLDs, indicating either that they have not been previously identified in embryonic stages or that they develop at a later time.

In mice, the V2 progenitor domain gives rise to a third class of neurons called V2c, which express Sox1 and only transiently Gata3 in very early development (prior to E12.5) before later downregulation (*Panayi et al., 2010*). It is unclear whether zebrafish have a homologous V2c population, although Sox1a/b is present in the 24 hpf spinal cord and notably also colabels with Gata3 (*Andrzejczuk et al., 2018*). A possible V2c homolog, referred to as V2s, has been identified as a Sox1a/b+ glycinergic cell type deriving from the V2 domain with long, ipsilateral, caudally projecting axons (*Gerber et al., 2019*) similar to the V2b-gly neurons described here. However, Gata3 expression was not investigated in V2s neurons, leaving it unclear whether V2s neurons are in fact V2b-gly (*Gerber et al., 2019*). Given the persistent, distinguishing expression of Gata3 in both V2b-gly and V2b-mixed subtypes, our data are consistent with the designation of two subclasses within V2b, not

a V2c homolog or additional V2s class. Further detailed investigation of Sox1a/b gene expression in these neurons will be required to clearly separate these classes.

## Speed specific inputs to motor circuits

Locomotion at faster versus slower speeds engages different sets of spinal interneurons, both within a genetically defined population (*McLean and Fetcho, 2009*; *McLean et al., 2008*; *Ampatzis et al., 2014*; *Zhong et al., 2011*) and across populations (*McLean et al., 2007*; *Talpalar et al., 2013*). Given the observation that slow motor neurons likely receive more input from V2b-mixed neurons whereas fast motor neurons receive largely V2b-gly input, it would be of interest to explore whether the V2b subpopulations are recruited at different speeds of locomotion. An alternative prediction is that, because V2b-gly and V2b-mixed somata extend over a shared DV range (*Figure 2D*), they are both likely to be recruited at over the same range of locomotor speeds, in line with the principle that DV position dictates speed-dependent recruitment of both motor neurons and interneurons (*Kimura and Higashijima, 2019*; *McLean et al., 2007*; *McLean et al., 2008*). Overall the specific postsynaptic consequences of glycinergic and mixed glycinergic/GABAergic inhibition remains unresolved. We posit that the fast decay kinetics of glycine-mediated transmission generates a restricted window of inhibition onto fast motor neurons, which undergo a faster recruitment cycle than slow motor neurons. In contrast, a combination of fast glycine and slower GABA-mediated currents might be expected to offer a longer window of postsynaptic current integration in slow motor neurons. In much of the nervous system, glycine receptor-mediated currents traditionally show faster decay rates than those from GABA receptors (*Awatramani et al., 2005*), including at interneuron inputs onto motor neurons (*Jonas et al., 1998*; *O'Brien and Berger, 1999*; *Russier et al., 2002*); with some exceptions (*Dumoulin et al., 2001*; *Eggers and Lukasiewicz, 2006*). Moreover, GABA can act as a coagonist on glycine receptors and in fact shorten the postsynpatic inhibitory response (*Lu et al., 2008*). Simultaneous whole cell recordings from V2bs and motor neurons will be necessary to elucidate the postsynaptic effects of V2b-gly versus V2b-mixed inputs.

Intra-V2b connectivity suggests a possible 'gear shift' within the V2bs, with the V2b-gly and V2b-mixed populations potentially inhibiting each other to enforce a given speed of swim. One caveat in interpretation of these results is that despite their different somatic positions, fast and slow motor neurons have overlapping dendritic fields (*Svara et al., 2018*). Therefore, it is possible that V2b-mixed neurons make synapses onto all motor neurons, and differential receptor expression is responsible for their IPSC pharmacology (*Figure 4*); meanwhile, V2b-gly might be responsible for other functions, such as suppression of dorsal horn sensory interneurons (*Li et al., 2004*). Paired recordings or higher resolution anatomical experiments will be required to distinguish these possibilities.

## V2b manipulation indicates role in speed control

One role of ipsilateral inhibition is to mediate flexor-extensor alternations via Ia reciprocal inhibition from V2b and V1 populations (*Britz et al., 2015*; *Zhang et al., 2014*). Ipsilaterally descending propriospinal neurons may also stabilize left-right alternation, although specific ablation of inhibitory ipsilaterally descending neurons has not been tested (*Ruder et al., 2016*; *Danner et al., 2017*). Our work establishes that selective activation of V2b neurons decreases tail beat frequency (*Figure 7*) while suppression leads to its increase (*Figure 6*). In contrast, genetic ablation of V1 neurons in mouse and larval zebrafish both led to a reduction in fictive locomotor speeds (*Gosgnach et al., 2006*; *Kimura and Higashijima, 2019*). In-phase V1 inhibition was found to limit slow motor neuron recruitment during fast movements through both direct and indirect (V1 onto V2a) synaptic pathways (*Kimura and Higashijima, 2019*). From this we surmise that ipsilateral inhibition from V1 and V2b shape distinct features of locomotion. V1 neurons may act to terminate motor neuron burst cycles whereas V2b neurons may limit overall speed, much like a brake.

The exact mechanism of V2b-associated speed changes remains unclear but is unlikely to be due to direct motor neuron spike shunting (*Figure 4*). One possible source of modulation is through local gating of excitatory drive on the dendrites, with V2b-mediated inhibition shunting coincident excitatory inputs during locomotion. Another possibility is that V2b neurons modulate motor circuitry via synapses onto other premotor targets, for example through V2b-V2a connectivity (M. Sengupta, data not shown), mirroring the V1 influence over motor circuitry via direct synapses onto both motor

neurons and V2a intermediaries (*Kimura and Higashijima, 2019*). These results point to a need for greater exploration of connectivity among spinal interneurons.

### Overall role of V2b in motor circuits

What is the functional role of V2b-mediated ipsilateral inhibition onto motor circuits? Three broad categories of V2b function occur. First, as discussed above, enforcing flexor-extensor and potentially forelimb-hindlimb alternation in limbed animals. Second, as supported in this work, V2b neurons may serve to titrate excitatory drive onto motor neurons differentially across varying speeds of movement. Measuring inhibitory conductances in motor neurons in vivo has revealed, surprisingly, that inhibition in-phase with excitation actually increases for increasingly strong movements (*Kishore et al., 2014*; *Berg et al., 2007*; *Berg et al., 2008*; *Li and Moult, 2012*), rather than diminishing to allow more powerful contractions. In this context, shunting ipsilateral inhibition might serve to enforce tight temporal control over spike timing via shortening membrane time constants. Surprisingly, ipsilateral inhibition does not appear to act as a source of gain control (*Figure 4*).

Thirdly, ipsilateral inhibition may act to isolate movements in certain behaviors that engage dedicated premotor circuitry, for example through the selective inhibition of interneurons during scratching versus swimming in turtle (*Berkowitz, 2002*; *Berkowitz, 2007*). Some V2b neurons, by virtue of their direct connections with ipsilateral motor circuits, could form part of the locomotor 'switch' from one behavior to another. A thorough investigation of V2b-gly and V2b-mixed recruitment during natural behaviors, such as speed transitions, turning, or balance, will allow us to better understand the similar or distinct ways that V2b subclasses influence locomotion.

## Materials and methods

### Key resources table

| Reagent type (species) or resource | Designation | Source or reference | Identifiers | Additional information |
|---|---|---|---|---|
| Genetic reagent (*Danio rerio*) | gata3:loxP-DsRed-loxP:GFP | This paper | ZFIN ID: ZDB-TG CONSTRCT-190724–1 | BAC line generation |
| Genetic reagent (*Danio rerio*) | gata3:Gal4 | This paper | ZFIN: ZDB-TG CONSTRCT-190724–2 | BAC line generation |
| Genetic reagent (*Danio rerio*) | gata3:ZipACR-YFP | This paper | ZFIN: ZDB-ALT-190813-3 | CRISPR knock-in |
| Genetic reagent (*Danio rerio*) | Gad1b:GFP | (*Satou et al., 2013*) | ZFIN: ZDB-ALT-131127–6 | BAC line generation |
| Genetic reagent (*Danio rerio*) | glyt2:loxP-DsRed-loxP:GFP | (*Kimura et al., 2014*) | ZFIN: ZDB-FISH-150901–22721 | CRISPR knock-in |
| Genetic reagent (*Danio rerio*) | UAS:CatCh | McLean lab (*Bagnall and McLean, 2014*) | | Plasmid and Tol-2 mediated DNA insertion |
| Software | WinWCP | J. Dempster, University of Strathclyde | RRID: SCR_014713 | |
| Software | Igor Pro procedure file | This paper | Github: https://github.com/bagnall-lab/V2b_paper_igor_code | |
| Software | Tail tracking Matlab code | This paper, adapted from *Severi et al. (2018)* | https://github.com/bagnall-lab/V2b_behavior | |

### Animal care

Adult zebrafish (*Danio rerio*) were maintained at 28.5˚C with a 14:10 light:dark cycle in the Washington University Zebrafish Facility following standard care procedures. Larval zebrafish, 4–7 days post fertilization (dpf), were kept in petri dishes in system water or housed with system water flow. Animals older than 7 dpf were fed rotifers daily. All procedures described in this work adhere to NIH guidelines and received approval by the Washington University Institutional Animal Care and Use Committee.

## Line generation

The *Tg(gata3:Gal4)* and *Tg(gata3:LoxP-dsRed-LoxP:GFP)* lines were generated via the bacterial artificial chromosome (BAC) transgenic technique (*Kimura et al., 2006*), using BAC zK257H17. The Gal4 and LRL-GFP constructs are described in *Kimura et al. (2013)* and *Satou et al. (2012)*, respectively. The *Tg(glyt2:LoxP-mCherry-LoxP:GFP)* line was generated with CRISPR/Cas9 genome targeting methods utilizing the short guide RNA, donor plasmid, and methods described in *Kimura et al. (2014)*. *Tg(gata3:zipACR-YFP)* animals were generated with CRISPR/Cas9 techniques using a gata3 short guide, TAG GTG CGA GCA TTG AGC TGA C. The donor Mbait-hs-zipACR-YFP plasmid was made by subcloning ZipACR (*Bergs et al., 2018*), obtained from Addgene, into a Mbait-hs-GFP plasmid with Gibson Assembly cloning methods. A UAS:CatCh (*Kleinlogel et al., 2011*) construct containing tol2 transposons was microinjected along with tol2-transposase RNA into one-cell *Tg (gata3:Gal4)* embryos to generate the *Tg(gata3:Gal4; UAS:CatCh)* line.

## Single-cell photoconversion

Fluorescent protein photoconversion was performed on anesthetized and embedded 5 dpf *Tg (gata3:Gal4; UAS:Kaede)* animals using an Olympus FV1200 microscope. Single-plane confocal images were continuously acquired to monitor conversion progress while 500 ms bursts of 405 nm light (100% intensity) were applied to an ROI ~1/10th the size of the targeted soma to elicit photoconversion. Animals were removed from agarose and allowed to recover in system water for 1–3 hr. After recovery, fish were anesthetized, embedded, and imaged as above. Tiled image stacks were acquired over an area ranging from the most rostral processes to the most caudal with a minimum of 10% area overlap between adjacent fields of view to aid the image stitching process.

## Single-cell dye electroporation

*Tg(gata3:LoxP-dsRed-LoxP:GFP; gad1b:GFP)* animals (5–6 dpf) were anesthetized in 0.02% MS-222 and three electroetched tungsten pins were placed through the notochord securing the animal to a Sylgard-lined 10 mm well dish. Forceps and an electroetched dissecting tool were used to remove skin and one segment of muscle fiber to expose the spinal cord. A pipette electrode filled with 10% Alexa Fluor 647 anionic dextran 10,000 MW (Invitrogen) in internal recording solution, was positioned to contact the soma of the target neuron. Dye was electroporated into the cell via one or more 500 ms, 100 Hz pulse trains (1 ms pulse width) at 2–5 V (A-M systems Isolated Pulse Stimulator Model 2100). Confocal imaging was performed as described above, after >20 min for dye filling.

## Fluorescent hybridization chain reaction (HCR)

Animals were fixed at 5 dpf in 4% paraformaldehyde and in situ hybridization was performed according to the HCR v3.0 protocol (*Choi et al., 2018*) with noted modifications. Preparation, dehydration and rehydration steps 1 through 14 were replaced with steps 2.1.1 through 2.2.8 with a Heat Induced Antigen Retrieval (HIAR) option in place of Proteinase K treatment (*King and Newmark, 2018*; *King and Newmark, 2013*). In situ probes were designed and distributed by Molecular Technologies (Beckman Institute, Caltech) to target gata3, gad1b, glyt2 (slc6a5), DsRed, mCherry, and GFP. Samples were kept in 4x saline-sodium citrate solution at 4°C prior to imaging. Samples were mounted in Vectashield (Vector Laboratories) or low-melting point agarose (Camplex SeaPlaque Agarose, 1.2% in system water) and positioned laterally on a microscope slides with #1.5 coverslip glass.

## Confocal imaging

5–7 dpf larvae were anesthetized in 0.02% MS-222 and embedded in low-melting point agarose in a 10 mm FluoroDish (WPI). Images were acquired on an Olympus FV1200 Confocal microscope equipped with high sensitivity GaAsP detectors (filter cubes FV12-MHBY and FV12-MHYR), and a XLUMPLFLN-W 20x/0.95 NA water immersion objective. A transmitted light image was obtained along with laser scanning fluorescent images. Sequential scanning was used for multi-wavelength images. Z-steps in 3D image stacks range from 0.8 to 1.4 microns. Fluorescent in situ hybridization samples were imaged with an UPLSAPO-S 30x/1.05 NA and silicone immersion media. Spectral images were collected for *Tg(gata3:zipACR-YFP; gata3:loxP-DsRed-loxP:GFP)* animals to distinguish between expression patterns of overlapping fluorophores. Samples were excited with a 515 nm

laser. Emission was collected with a PMT detector from 10 nm wide spectral windows across the emission range 525–625 nm for each z-plane. Spectral deconvolution was performed with Olympus Fluoview software.

## Electrophysiology

Whole-cell patch-clamp recordings were targeted to V2bs or motor neurons in *Tg(gata3:Gal4; UAS: CatCh), Tg(gata3:zipACR-YFP)*, doubly-transgenic *Tg(gata3:LoxP-dsRed-LoxP:GFP; gad1b:GFP) or Tg(gata3:zipACR-YFP; gata3:loxP-DsRed-loxP:GFP)* larvae at 4–6 dpf. Larvae were immobilized with 0.1% $\alpha$-bungarotoxin and fixed to a Sylgard lined petri dish with custom-sharpened tungsten pins. One muscle segment overlaying the spinal cord was removed at the mid-body level (segments 9–13). The larva was then transferred to a microscope (Scientifica SliceScope Pro or Nikon Eclipse) equipped with infrared differential interference contrast optics, epifluorescence, and immersion objectives (Olympus: 40X, 0.8 NA; Nikon: 60X, 1.0 NA). The bath solution consisted of (in mM): 134 NaCl, 2.9 KCl, 1.2 $MgCl_2$, 10 HEPES, 10 glucose, 2.1 $CaCl_2$. Osmolarity was adjusted to ~295 mOsm and pH to 7.5.

Patch pipettes (5–15 M$\Omega$) were filled with internal solution for current clamp composed of (in mM): 125 K gluconate, 2 $MgCl_2$, 4 KCl, 10 HEPES, 10 EGTA, and 4 $Na_2ATP$; for voltage clamp, 122 cesium methanesulfonate, one tetraethylammonium-Cl, 3 $MgCl_2$, 1 QX-314 Cl, 10 HEPES, 10 EGTA, and 4 $Na_2ATP$. Additionally, Alexa Fluor 568 or 647 hydrazide 0.05–0.1 mM, or Sulforhodamine 0.02% was included. Osmolarity was adjusted to ~285 mOsm and KOH or CsOH, respectively was used to bring the pH to 7.5. Peripheral nerve recordings were made with flame polished pipettes with a ~ 30–50 μm diameter tip, filled with bath solution. Patch recordings were made in whole-cell configuration using a Multiclamp 700B, filtered at 10 kHz (current clamp) or 2 kHz (voltage clamp.) Peripheral nerve activity was recorded in fixed current configuration, signal was processed with 100x gain with 200 Hz and 2 kHz filtering. All recordings were digitized at 20 kHz with a Digidata 1440 or 1550 (Molecular Devices) and acquired with WinWCP (J. Dempster, University of Strathclyde) or pCLAMP 10 (Molecular Devices).

The *Tg(gata3:Gal4; UAS:CatCh)* line labels both V2b and Kolmer-Agduhr/cerebrospinal fluid-contacting neurons (CSF-cNs). To ensure that evoked IPSCs derived from presynaptic V2bs rather than CSF-cNs, epifluorescent illumination was targeted 3–7 segments rostral to the recorded segment. A Polygon400 Digital Micromirror Device (Mightex) was used to provide patterned illumination in indicated recordings. Previous studies found that CSF-cNs have short ascending axons and do not contact any motor neurons besides the caudal primary (CaP) (Hubbard et al., 2016). CatCh expression in the *Tg(gata3:Gal4; UAS:CatCh)* line is variegated, with some animals showing strong CatCh expression throughout both CSF-cNs and V2bs, and others showing good expression in CSF-cNs and minimal expression in V2b cells. Additional control experiments were performed in animals with minimal V2b label to demonstrate the absence of contribution of CSF-cN synapses in these experiments. Strychnine and gabazine (SR-95531) were applied at 10 μM.

In current clamp experiments examining V2b effects on motor neuron firing, 30 ms depolarizing current steps were delivered to motor neurons in increasing amplitude steps (primary motor neurons: 25 pA steps; secondary motor neurons: 5–25 pA steps), alternating between delivery with and without V2b activation targeted 3–5 segments rostral to the recording. V2b neurons were activated with a 10–20 ms pulse of blue light, delivered either via epifluorescence of via the Polygon DMD, as above, that initiated 10 ms prior to the onset of the depolarizing step in order to maximize the peak of V2b inhibition during the current step. At resting membrane potentials, a modest (1–5 mV) non-specific depolarizing current was often seen in response to this optogenetic stimulation; it was insensitive to strychnine, gabazine, and two blockers of serotonergic receptors (cyproheptadine and mianserin, both at 1 μM; data not shown), and therefore seems likely to arise from gap junctional networks of unclear identity. To prevent this from interfering with recordings, small squares of illumination were delivered via DMD to target just V2b neurons, or epifluorescence apertures were narrowed to the smallest possible, minimizing the depolarizing current. As a positive control for the efficacy of this protocol in driving V2b-mediated inhibition, we found that spiking in V2a neurons was significantly reduced by V2b optogenetic activation (data not shown), indicating that the absence of effect on motor neuron spiking was not due to a failure of V2b activation.

Motor neurons were identified by axon fill that extended into the musculature and/or by retrograde dye labeling from the muscle. For retrograde labeling, 4 dpf larvae were anesthetized (0.02%

MS-222) and laid flat on an agarose plate. A Narishige micromanipulator in conjunction with a micro-injection pump (WPI, Pneumatic Picopump) was used to deliver small volumes of dye (Alexa Fluor 568 dextran, 3000 MW) via glass pipette into the muscle. Fish recovered in regular system water and were subsequently used for recordings at 5–6 dpf.

Data were imported into Igor Pro using NeuroMatic (*Rothman and Silver, 2018*). Spike threshold was defined as 10 V/s, and custom code was written to determine spike width and afterhyperpolarization of the initial spike elicited by pulse steps, code available on Github (*Bagnall-lab, 2019a*; copy archived at https://github.com/elifesciences-publications/V2b_paper_igor_code). Input resistance was calculated by an average of small hyperpolarizing pulses. To isolate IPSCs, 10 μm NBQX was present in the bath and neurons were voltage clamped at the EPSC reversal potential.

Motor neurons at the dorsal extent of the distribution (>50% of distance from bottom of spinal cord to top) exhibited lower input resistances (mean ± SD: 287 ± 75 MΩ) and were considered 'fast' and the remainder, which exhibited higher input resistances (885 ± 367 MΩ) considered 'slow' (*McLean et al., 2007*). These groups mostly correspond to primary and secondary motor neurons, but some dorsally located bifurcating secondaries may be included in the fast group (*Menelaou and McLean, 2012*).

Optogenetic validation of ZipACR in V2b and CSF-cN was performed on *Tg(gata3:zipACR-YFP)* and *Tg(gata3:zipACR-YFP; gata3:loxP-DsRed-loxP:GFP)* animals. Light stimulation was provided with high intensity epifluorescent illumination (CoolLED pE-300), 5–10% intensity with a 40X (0.8 NA) water-immersion objective, and low intensity illumination which is identical to the conditions of behavioral recordings, 100% intensity with a 4X (0.1 NA) air objective.

## Image analysis

Image analysis was performed with ImageJ (FIJI) (*Schindelin et al., 2012*). Igor Pro 6 was utilized for data analysis and statistics unless otherwise noted. V2b cell counts and neurotransmitter coexpression was quantified manually by two researchers (R.C. and M.J.); no significant differences in quantification were detected. Gata3+ V2b cells were identified and marked (ImageJ Cell Counter) relative to spinal cord and segment boundaries, giving total V2b/segment quantities. Subsequently, each cell was evaluated for expression of fluorescent proteins marking Gad1b or Glyt2.

Transgenic line validation was performed with in situ hybridization and quantified by two researchers (M.B. and R.C.) with no significant discrepancy in results. A ~ 3–5 μm z-stack projection was made in a cell-dense area of spinal cord spanning two to three segments for each animal. ROIs of neurons were drawn in one channel before checking whether there was colocalization in the other channel. Samples were quantified twice: once for completeness (percentage of endogenous RNA positive neurons also expressing the transgene) and once for accuracy (percentage of transgene labeled neurons positive for endogenous RNA). 3–7 animals were evaluated in each line.

For axon tracing, stitched projection images were made with the Pairwise stitching (*Preibisch et al., 2009*) ImageJ plugin. The overlap of the fused image was smoothed with linear blending and was registered based on the fill channel or the average of all channels. Photoconversion cell fill images underwent an extra processing step in which the bleached green channel was subtracted from the photoconverted red channel. The Simple Neurite Tracer plugin (*Longair et al., 2011*) was used to trace the axon projection and branching relative to marked spinal cord boundaries. Axon lengths are reported as the number of segments transversed.

Motor neuron dendrites were quantified from confocal z-stack images of *Tg(mnx:GFP)* 5 dpf animals. Images were cropped to a single hemisegment. The Weka Trainable Segmentation plugin (*Arganda-Carreras et al., 2017*) was used to segment the motor neuron image into three classifiers; somata, axons exiting the spinal cord, and dendrites. Classification was based on Hessian training features. Training was performed iteratively for each image. The binary segmented images were applied to mask all non-dendrite fluorescence (n = 4 hemisegments/animal; n = 3 animals.) Fluorescence was maximum intensity projected in the z-dimension, collapsed along the horizontal plane and normalized to give an estimate of motor neuron dendrite density in the dorsoventral plane of the spinal cord.

## NMDA-induced behavior and ZipACR optogenetic supression

5 dpf *Tg(gata3:zipACR-YFP)* animals and clutchmates were anesthetized in 0.02% MS-222 and placed on an agar plate under a dissecting microscope. A complete spinal cord transection was made with Vannas spring microscissors, plus a sharpened pin if necessary, between spinal cord segments 2 and 5. Tail blood flow was monitored post-transection and throughout the preparation; animals with significantly reduced blood flow were euthanized and not used for recordings. After transection the animal briefly recovered in extracellular solution and then was embedded in a dorsal up position in 1.2% low melting point agarose. Solidified agarose surrounding the tail caudal to the transection was removed with a dissection scalpel. 200 µM NMDA (Sigma Aldrich) in extracellular solution was added to the dish. Recordings were initiated after tail movement began, typically 2–10 min later.

Behavior experiments were performed with a Scientifica SliceScope upright microscope equipped with a Fastec HiSpec1 camera and an Olympus Plan N 4x/0.10 objective. Image collection was made with Fastec acquisition software. Images were acquired at 200 Hz for 5 s. Optical stimulation was made with 100% intensity full field epi-illumination from a CoolLED pe300ultra source routed through a GFP filter cube (Chroma 49002). Recordings with optical stimulation were alternated with recordings without stimulation; n = 6–17 recordings for each animal.

Analysis was run in MATLAB R2017a with custom code adapted from *Severi et al. (2018)*, available for download from Github (*Bagnall-lab, 2019b*; copy archived at https://github.com/elifescien-ces-publications/V2b_behavior). The caudal edge of the transection and the tail periphery were manually selected as tail boundaries and 10 points for tracking were evenly distributed along the body. The caudal-most tail point was used to calculate tail speed (mm/s) at each frame of the recordings. A tail speed threshold of 0.5 mm/s was used to distinguish true movement from tail drift. Tail movement amplitude was calculated as the maximum tail displacement in the initial second of each recording. Tail beat frequency was computed from left-to-right tail oscillations during manually identified movement bouts; 6–30 consecutive peaks were averaged for each recording.

## Evoked swim with optogenetic activation

5–6 dpf *Tg(gata3:Gal4; UAS:CatCh)* and sibling animals were paralyzed and pinned to Sylgard dishes with three sharpened Tungsten pins. 3–4 segments of skin were removed and the recording electrode was placed on the body near the intermyotomal cleft. Swim was induced with a bipolar Tungsten electrode placed on an intact portion of skin ~5 segments caudal to the recording site with an empirically determined stimulation level set to reliably evoke swims (5–50 V). Optogenetic stimulation was made with epifluorescent light through standard filter cubes and a 40x objective at 5% light intensity with 20 ms on/off light cycles under the following conditions: no light, GFP excitation, and RFP excitation. Electical stimulation occured 100 ms post light stimulation. Trials in which the animal initiated swim prior to the electrical stimulation were excluded from analysis. Optical stimulation was made 3–5 segments rostral and 3–5 segments caudal to the peripheral motor recording to assess the contributions of descending (V2b) and ascending (CSF-cN) inhibition to the motor activity. Each recording paradigm was repeated for 16–32 trials. Peripheral motor nerve signal was rectified and inter-beat interval frequencies were manually tabulated in Matlab by experimenter blind to treatment conditions. Trial averages are reported for each recording condition in each animal.

### Resource sharing

Transgenic fish lines generated in this work are available from the authors upon request. Example data and statistics for soma quantification, V2b cell traces, motor neuron dendrite mapping, whole cell electrophysiology and optogenetic connectivity, behavior experiments, and example images are available for download from Dryad (DOI: https://doi.org/10.5061/dryad.1d78mt2). Custom written code for spike analysis and tail tracking can be downloaded from Github (https://github.com/bag-nall-lab/).

## Acknowledgements

We are grateful to Drs. David McLean and Sandeep Kishore for sharing the *Tg(UAS:Kaede)* and *Tg (gad1b:GFP)* transgenic lines and the UAS:Catch plasmid. Thanks to Dr. Kristen Severi for providing

tail tracking code, Marquise Jones for cell counting work, and Dr. Paul Stein for his thoughtful insights. We gratefully acknowledge the Washington University Zebrafish Facility. Imaging experiments were performed in part through the use of Washington University Center for Cellular Imaging (WUCCI) supported by Washington University School of Medicine, The Children's Discovery Institute of Washington University and St. Louis Children's Hospital and the Foundation for Barnes-Jewish Hospital. This work was supported by funding through the National Institute of Health (NIH) R00 DC012536 (MWB), R01 DC016413 (MWB), a Sloan Research Fellowship (MWB), The Children's Discovery Institute of Washington University (MWB), NINDS F32 NS103247 (RAC), and the National Bio-Resource Project in Japan (SH). MWB is a Pew Biomedical Scholar and a McKnight Foundation Scholar.

## Additional information

### Funding

| Funder | Grant reference number | Author |
| --- | --- | --- |
| National BioResource Project | | Shin-ichi Higashijima |
| National Institute on Deafness and Other Communication Disorders | R00 DC012536 | Martha W Bagnall |
| National Institute on Deafness and Other Communication Disorders | R01 DC016413 | Martha W Bagnall |
| National Institute of Neurological Disorders and Stroke | F32 NS103247 | Rebecca A Callahan |
| Alfred P. Sloan Foundation | | Martha W Bagnall |
| Pew Charitable Trusts | | Martha W Bagnall |
| McKnight Endowment Fund for Neuroscience | | Martha W Bagnall |
| Children's Discovery Institute | | Martha W Bagnall |

The funders had no role in study design, data collection and interpretation, or the decision to submit the work for publication.

### Author contributions

Rebecca A Callahan, Conceptualization, Formal analysis, Funding acquisition, Investigation, Methodology, Writing—original draft, Project administration, Writing—review and editing; Richard Roberts, Investigation; Mohini Sengupta, Investigation, Methodology; Yukiko Kimura, Shin-ichi Higashijima, Resources, Methodology; Martha W Bagnall, Conceptualization, Resources, Formal analysis, Supervision, Funding acquisition, Investigation, Methodology, Writing—original draft, Project administration, Writing—review and editing

### Author ORCIDs

Mohini Sengupta [ID] https://orcid.org/0000-0002-5234-8258
Yukiko Kimura [ID] https://orcid.org/0000-0001-8381-8622
Shin-ichi Higashijima [ID] https://orcid.org/0000-0001-6350-4992
Martha W Bagnall [ID] https://orcid.org/0000-0003-2102-6165

### Ethics

Animal experimentation: This research adheres to recommendations in the Guide for the Care and Use of Laboratory Animals of the National Institutes of Health and received approval by the Washington University Institutional Animal Care and Use Committee (protocol 20170228).

Decision letter and Author response
Decision letter https://doi.org/10.7554/eLife.47837.027
Author response https://doi.org/10.7554/eLife.47837.028

## Additional files

### Supplementary files

• Transparent reporting form
DOI: https://doi.org/10.7554/eLife.47837.022

### Data availability

Datasets have been deposited on Dryad: https://doi.org/10.5061/dryad.1d78mt2.

The following dataset was generated:

| Author(s) | Year | Dataset title | Dataset URL | Database and Identifier |
|---|---|---|---|---|
| Callahan RA, Roberts R, Sengupta M, Kimura Y, Higashijima SI, Bagnall MW | 2019 | Data from: Spinal V2b neurons reveal a role for ipsilateral inhibition in speed control | https://dx.doi.org/10.5061/dryad.1d78mt2 | Dryad, 10.5061/dryad.1d78mt2 |

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

# Appendix 1

DOI: https://doi.org/10.7554/eLife.47837.023

## Supplementary Information

In situ hybridization validates transgenic animal lines

To validate the transgenic lines used in this work, we performed two-color fluorescent in-situ hybridization on each line, examining whether the fluorescent reporter expression matched with RNA expression of the targeted gene for *Tg(gata3:loxP-DsRed-loxP:GFP)*, *Tg (gata3:Gal4; UAS:GFP)*, *Tg(gad1b:GFP)*, and *Tg(glyt2:loxP-mCherry-loxP:GFP)*. Results for the completeness and accuracy of transgenic lines are reported in *Table 1* and example images are provided in *Figure 1—figure supplement 1*. All lines were sufficiently complete and accurate for use in further quantitative analyses.

Serotonergic Gata3+ cells

Gata3 is expressed in a third class of neurons located along the ventral edge of the spinal cord. These rectangular neurons did not colabel with markers for glycinergic, GBABergix, or glutamatergic identity in *Tg(glyt2:loxP-mCherry-loxP:GFP)*, *Tg(gad1b:GFP)*, or *Tg(vglut2a: Gal4; UAS:GFP)* lines, respectively (data not shown). Single cell axon fills in *Tg(gata3:Gal4; UAS:Kaede)* reveal neural morphology very similar to that of intraspinal serotonergic neurons (ISN) described by *Montgomery et al. (2018)* including short cell processes that project rostrally and caudally from the soma and a stereotypical growth cone present on the caudal process (*Figure 1—figure supplement 2*). Given the distinct morphology and lack of neurotransmitter expression we conclude the ventral Gata3+ neurons are likely ISNs. Serotonergic neurons in the hindbrain express Gata2/Gata3 (*Craven et al., 2004*; *van Doorninck et al., 1999*) during development and a subset of Gata3+ CSF-cN neurons express 5-HT early in development (<3 dpf) (*Montgomery et al., 2016*). On average 3.05 + /- 1.19 (mean + /- SD) ISN are labeled per segment in *Tg(gata3:loxP-DsRed-loxP:GFP)* and *Tg (gata3:ZipACR-YFP)* lines consistent with cell densities previously described for the ISN population. Only a small subset of ISNs, 0.75 + /- 0.68 (mean + /- SD) expressed fluorescent protein in *Tg(gata3:Gal4; UAS:CatCh)* animals. Activation of ISN has been shown to reduce the incidence of NMDA-induced swim bouts in transected preparations (*Montgomery et al., 2018*); however no effect on swim rhythm or speed was reported. Therefore, it is unlikely that ISN activation, in CatCh experiments with minimal ISN expression, or suppression, in ZipACR experiments, led to the observed tail beat frequency changes.

