## [Decision Letter]

Thank you for submitting your article "Spinal V2b neurons reveal a role for ipsilateral inhibition in speed control" for consideration by *eLife*. Your article has been reviewed by three peer reviewers, including Claire Wyart as the Reviewing Editor and Reviewer #1, and the evaluation has been overseen by Ronald Calabrese as the Senior Editor. The following individual involved in review of your submission has agreed to reveal their identity: Marco Beato (Reviewer #2).

The reviewers have discussed the reviews with one another and the Reviewing Editor has drafted this decision to help you prepare a revised submission.

Summary:

Callahan et al. present evidence that in the zebrafish larval spinal cord spinal V2b inhibitory interneurons control locomotor speed. This is a very exciting finding as V2b are ipsilateral, descending, and inhibitory interneurons. This finding, along with the recent publication on V1 neurons involved in enhancing speed (Kimura and Higashijima, 2019), provide more evidence that speed control is distributed across many interneurons in the spinal cord and not only restricted to excitatory interneurons.

Their approach relies on novel BAC transgenic lines where gata3 drives the expression in V2b neurons, together with cerebrospinal fluid-contacting neurons (CSF-cNs, ipsilateral ascending neurons originating from p3 and pMN). However, since in contrast, CSF-cNs innately increase locomotor speed during the escape (Bohm et al., 2016), the increase in locomotor speed observed when suppressing activity of V2b and CSF-cNs together should be due to V2b intrinsically reducing locomotor speed.

Furthermore, the manuscript provides proof of anatomical segregation of the axons of two distinct classes of V2b interneurons, one purely glycinergic and one with mixed GABAergic-Glycinergic phenotype and indicates some novel rules of connectivity, with fast MNs receiving mostly glycinergic inhibition and slow motoneurons receiving mixed inhibitory input. Optogenetic silencing of both classes of V2b neurons results in increased tail beat frequency, leading to the hypothesis that V2b neurons might have a role in controlling locomotor speed. The methodology is sound, all results are clearly represented and the major findings are fully supported by the data. While a role of V2b interneurons in controlling the speed might not come as a complete surprise, the authors give a first functional proof of direct connectivity of V2b and motoneurons, as well as (rather more surprisingly) of connectivity between V2b interneurons themselves.

As raised by two of the reviewers, the ultimate goal would be to be able to activate/silence the two classes of V2b separately, as well as establishing their rules of connectivity between themselves. Such experiments would require complex intersectional genetics, which would take time and therefore would be better suited for a separate study.

V2b interneurons appear to act as a break on the central pattern generator (CPG), but it is not clear how they do so. In order to specify the underlying mechanisms, it would be interesting to add to this study two experiments that should be doable in two months:

1) Optogenetic activation of V2b using an excitatory opsin in the *Tg(gata3:gal4)* line during locomotion to test whether it would slow down locomotion:

While the authors show the neat result of increased tail beat frequency upon silencing of V2b interneurons, they do not show what happens when such interneurons are activated during tail beat. This would be a relatively simple experiment to perform and would add some information on the potential function of V2b interneurons.

2) Recordings of MNs in V-Clamp and C-Clamp with optogenetic control of V2b to elicit single spike in order to resolve the properties of the V2b-MN synapses and test how gaba+ versus gaba- V2b firing impact the firing of slow versus fast MNs:

By increasing the selectivity of the light stimulation to single neuron/single spike (using short pulses of 5ms instead of 20-50ms as previously done), the authors could calibrate the light to get 'unitary' responses in motoneurons, evoke firing in motoneurons (by current injection) and see how activation of one or a group of V2b would alter the firing. Following that, the authors could verify whether the input is mixed or purely glycinergic (and onto fast or slow motoneurons). This should be quite elegant, informative and not too complicated.

With these two experiments added, this study should lay solid basis for future studies aimed at understanding how the differential connectivity onto fast and slow motoneurons of mixed and pure glycinergic V2b interneurons might affect the function of the locomotor network. The finding of a recurrent disinhibitory loop between V2bs raises interesting possibility of complex interactions and modulations within the premotor network.

Specific comments:

1) All V2b are glycinergic but some are also GABAergic – overall slightly more ventral with more ventral projections. This mixed neurotransmitter release should slow down IPSCs on their targets. Accordingly, there is a correlate with the projection from V2b onto slow and fast types of MNs. How does the fast glycinergic inhibition impact the firing of fast motoneurons? and reciprocally the slow GABAergic inhibition the firing of slow motoneurons? By doing so, can the authors speculate on how these ipsilateral ascending neurons can slow down the rhythm?

2) Summary:

Specify more how V2b project onto different motoneuron (MN) types ("differential targeting to slower and faster circuits"). It would be nice to go further by presenting how the projections and neurotransmitter types together with circuit disinhibition can enable the modulation of speed observed.

3) Figure 1B: What are the ventral-most cells in this panel (2-3 per segment)? They appear in the V3 domain but do not look like CSF-cNs on the image, could V3 or V2c interneurons also be in the line?

4) Filled cells in Figure 4 refer to segments 14-18. However, according to the data in Figure 1E, those are among the shortest axon (shorter that the 10-11 group), so I am wondering whether the same neat dorsoventral separation is also seen in higher level segments, especially for the V2b with longest axons (segments 10-11, judging from Figure 1E).

5) Figure 4, use same Y-scale for F,G and H.

6) Please report concentrations of strychnine and gabazine. None of the two drugs is selective. While there is no doubt on the qualitative results of Figure 5, knowing the concentrations would allow to judge them quantitatively. Also, how was the extent of block measured? The text reports conductances, so I would assume that the peak was measured. However, since the responses in Figure 5 and 6 originate from multiple spikes in multiple pre-synaptic neurons, the integral of the trace might be more appropriate.

7) Figure 7: Light activation seems to depolarize the cell (opposite to what shown in Figure 6—figure supplement 2). It is highly possible that chloride gradients vary among cells and that minor change in resting potential could change the effect. Is the difference observed between Figure 7 and Figure 6—figure supplement 2 coming from different resting potentials in the two cells shown? In both cases the increased anion conductance causes a large shunt, so there is no doubt about the silencing of neurons. Still, it would be better to have a label with the resting membrane potential.

8) The spikes represented in Figures 6E and 7B are rather small (10-15 mV amplitude). Overall, it seems that action potentials in V2b interneurons do not have a classical shape, while AP in CSF-cN neurons recorded in the same conditions do (Figure 7 and Figure 6—figure supplement 2). Are these representative recordings? Is it a specific feature of these neurons? Can you comment on that?

9) Introduction, paragraph two: the statement that premotor neurons only belong to five superclasses is not 100% correct. Of the 10 subfamily, at least Di3 have been shown to contact motoneurons (Bui et al., 2013).

10) Consider moving the 'in situ hybridization' section, table and figure to the Materials and methods or supplementary information. It is a necessary control, but interrupts the flow of information while reading the paper.

11) Results, second paragraph, and Figure 1: I was wondering whether illustrating the position of the soma of V2b interneurons with a coronal view would be more readable. Indeed, I found the use of false color depth-coded image in panel B difficult to read. Is the red background corresponding to the other half of the spinal cord? Because the color scale states L to R (supposedly left to right), while in the text it is written from medial to lateral. Showing a projection for a full segment for example on a coronal slice would help the reader.

12) Subsection “Neurotransmitter expression defines subpopulations of V2b neurons”: would it be possible to provide data showing the expression of both GABA and glycine in the same neuron? Are purely GABAergic neurons existing? What is the functional consequences of having a co-release of GABA and glycine?

13) Figure 7: the panel E is not easy to read and as such does not provide any obvious info. Is this panel really necessary?

14) Please correct the opening sentence in subsection “V2b axons extend throughout the spinal cord” (probably a mix between two version of the same sentence).

15) Paragraph three subsection “Neurotransmitter expression defines subpopulations of V2b neurons”: please correct sunclasses

16) Subsection “Axonal morphology varies by subpopulation identity”: please define VeLD (this is done in the Discussion but not here)

17) Figure 6—figure supplement 2: panel A the time scale on top might be wrong as the stim duration is 20ms while the scale indicates 0.4 sec?

---

## [Author Response]

In the original manuscript, we suppressed V2b activity and saw that locomotion sped up; reviewers suggested activating V2b neurons as a complementary experiment. We carried out this experiment using optogenetic activation of V2b neurons in a fictive swim preparation and indeed, saw that locomotion slowed down. These new data, along with several controls to address possible confounding influences, are now shown in revised Figure 7. This result significantly supports our conclusion that V2b neurons can serve as a “brake” on locomotor activity.

In the original manuscript, we used voltage clamp to show that V2b neurons evoked IPSCs in fast and slow motor neurons; reviewers suggested evaluating the effects on motor neuron spiking in current clamp. We carried out these experiments by pairing short depolarizing steps in motor neurons with optogenetic activation of V2b neurons and were rather surprised by the result: on average, there was no effect on threshold or evoked spiking. We did see some motor neurons (3/14 fast motor neurons) with a clear increase in threshold and rightward shift of the input-firing curve, but on average there was no effect. We show these new data in Figure 4F and speculate that V2b-mediated IPSCs may occur on the dendrites, in line with anatomical data about the axon position in the white matter, and therefore might affect dendritic integration but not somatically-evoked firing. We also mention in the Discussion that it is possible the behavioral effects of V2b manipulation are partially or primarily mediated through inhibitory effects onto other spinal neurons, such as the V2a class (for which we have preliminary evidence, described in Materials and methods).

V2b interneurons appear to act as a break on the central pattern generator (CPG), but it is not clear how they do so. In order to specify the underlying mechanisms, it would be interesting to add to this study two experiments that should be doable in two months:1) Optogenetic activation of V2b using an excitatory opsin in the Tg(gata3:gal4) line during locomotion to test whether it would slow down locomotion:While the authors show the neat result of increased tail beat frequency upon silencing of V2b interneurons, they do not show what happens when such interneurons are activated during tail beat. This would be a relatively simple experiment to perform and would add some information on the potential function of V2b interneurons.

We have now carried out this experiment, and indeed activation of V2b neurons slows locomotion, consistent with the notion that they serve at least in part as a brake. These new results are shown in revised Figure 7.

2) Recordings of MNs in V-Clamp and C-Clamp with optogenetic control of V2b to elicit single spike in order to resolve the properties of the V2b-MN synapses and test how gaba+ versus gaba- V2b firing impact the firing of slow versus fast MNs:By increasing the selectivity of the light stimulation to single neuron/single spike (using short pulses of 5ms instead of 20-50ms as previously done), the authors could calibrate the light to get 'unitary' responses in motoneurons, evoke firing in motoneurons (by current injection) and see how activation of one or a group of V2b would alter the firing. Following that, the authors could verify whether the input is mixed or purely glycinergic (and onto fast or slow motoneurons). This should be quite elegant, informative and not too complicated.With these two experiments added, this study should lay solid basis for future studies aimed at understanding how the differential connectivity onto fast and slow motoneurons of mixed and pure glycinergic V2b interneurons might affect the function of the locomotor network. The finding of a recurrent disinhibitory loop between V2bs raises interesting possibility of complex interactions and modulations within the premotor network.Specific comments:1) All V2b are glycinergic but some are also GABAergic – overall slightly more ventral with more ventral projections. This mixed neurotransmitter release should slow down IPSCs on their targets. Accordingly, there is a correlate with the projection from V2b onto slow and fast types of MNs. How does the fast glycinergic inhibition impact the firing of fast motoneurons? and reciprocally the slow GABAergic inhibition the firing of slow motoneurons? By doing so, can the authors speculate on how these ipsilateral ascending neurons can slow down the rhythm?

We have carried out this experiment as requested, and interestingly do not see on average any effect of V2b-mediated inhibition on somatically-evoked spiking. These results, which may reflect dendritically positioned V2b-MN synapses, are shown in Figure 4F. We speculate that dendritic integration in motor neurons may be affected by V2b inhibition, and/or that the major effects of V2b on the circuit are mediated through influence on V2a neurons (Discussion).

2) Summary:Specify more how V2b project onto different motoneuron (MN) types ("differential targeting to slower and faster circuits"). It would be nice to go further by presenting how the projections and neurotransmitter types together with circuit disinhibition can enable the modulation of speed observed.

We have now addressed the question of glycinergic vs. GABAergic kinetics of inhibition in the Discussion. We think there are two plausible hypotheses: (1) Since glycinergic kinetics are typically faster than GABAergic, glycinergic targeting to fast MNs may be “matched” to the faster membrane time constants and oscillations that these neurons must produce, and vice versa with GABA; or (2) Since V2b-gly and V2b-mixed somata occupy roughly the same dorsal-ventral territory, and DV soma position is highly correlated with swim speed recruitment in other zebrafish spinal neurons (e.g. McLean et al., 2007), both the V2b-gly and V2b-mixed subtypes are recruited for a range of swim speeds. Clearly these interesting and conflicting hypotheses will be tested with future experiments!

3) Figure 1B: What are the ventral-most cells in this panel (2-3 per segment)? They appear in the V3 domain but do not look like CSF-cNs on the image, could V3 or V2c interneurons also be in the line?

Thank you for pointing out these additional ventrally positioned cells. We have now anatomically identified them as intraspinal serotonergic neurons (ISN), using a cell fill approach shown in new Figure 1—figure supplement 2. We have also added text to further describe them in subsection “Gata3 transgenic lines label V2b neurons” paragraph two as well as in a supplementary text section; additionally, they are now identified in Figure 1. Given their presence in *Tg(gata3:ZipACR-YFP)* and *Tg(gata3:loxP-dsRed-loxP:GFP)* animals (though not *Tg(gata3:Gal4; UAS:CatCh)*) we were initially concerned about any influence that ISN activation may have on the behavioral experiments. However, published work on ISN activation and bath serotonin application demonstrated that serotonin has no influence over tail beat frequency in larval zebrafish, and thus is unlikely to skew our behavioral results showing V2b’s influence over tail beat patterning. We further confirmed this absence of effect with Jacob Montgomery and Mark Masino, who have recently published on ISN in larval zebrafish.

4) Filled cells in Figure 4 refer to segments 14-18. However, according to the data in Figure 1E, those are among the shortest axon (shorter that the 10-11 group), so I am wondering whether the same neat dorsoventral separation is also seen in higher level segments, especially for the V2b with longest axons (segments 10-11, judging from Figure 1E).

Thank you for this comment; it illustrates a need for clarity in the presentation of our data in Figure 1E. In this figure, we found that cells originating in segments 10-11 extend axons that are statistically longer than those originating in segments 1-3 and segments 20 through the caudal end of the spinal cord. We have adjusted the description to make this clearer in paragraph two of subsection “V2b axons extend throughout the spinal cord”. However, the point you make regarding whether the DV relationship persists along the extent of the spinal cord is intriguing. We did observe axon trajectories similar to both the V2b-gly and V2b-mixed pathways, i.e. dorsally and ventrally positioned respectively, from Kaede photoconversions which were carried out along the full rostrocaudal axis of the spinal cord. However, we don’t have neurotransmitter identity from those experiments (because Kaede occupies both green and red fluorescence channels) so can’t state with confidence that these correspond to the V2b-gly and -mixed subtypes.

5) Figure 4, use same Y-scale for F,G and H.

Figure 4 has been modified as suggested.

6) Please report concentrations of strychnine and gabazine. None of the two drugs is selective. While there is no doubt on the qualitative results of Figure 5, knowing the concentrations would allow to judge them quantitatively. Also, how was the extent of block measured? The text reports conductances, so I would assume that the peak was measured. However, since the responses in Figure 5 and 6 originate from multiple spikes in multiple pre-synaptic neurons, the integral of the trace might be more appropriate.

Concentrations are now reported in Materials and methods subsection “Electrophysiology” (10 µM for each). Regarding quantification of the block, we saw similar results both with peak and integral measurements, but the integral of the trace yielded more variable results depending on the duration of the integration window. However, the results holds up regardless of the choice of integral duration, and we now report the results from a 100 ms window in the text.

7) Figure 7: Light activation seems to depolarize the cell (opposite to what shown in Figure 6—figure supplement 2). It is highly possible that chloride gradients vary among cells and that minor change in resting potential could change the effect. Is the difference observed between Figure 7 and Figure 6—figure supplement 2 coming from different resting potentials in the two cells shown? In both cases the increased anion conductance causes a large shunt, so there is no doubt about the silencing of neurons. Still, it would be better to have a label with the resting membrane potential.

Thank you for this point; the reviewer is correct that this cell happened to be resting quite hyperpolarized, so that the chloride conductance was depolarizing (though shunting). We have added membrane potential labels as suggested, but we also replaced that example neuron with another that was resting closer to Erev(Cl) to reduce confusion among less experienced readers of the manuscript.

8) The spikes represented in Figures 6E and 7B are rather small (10-15 mV amplitude). Overall, it seems that action potentials in V2b interneurons do not have a classical shape, while AP in CSF-cN neurons recorded in the same conditions do (Figure 7 and Figure 6—figure supplement 2). Are these representative recordings? Is it a specific feature of these neurons? Can you comment on that?

Yes, this is an interesting observation. In some classes of zebrafish spinal neurons it is quite common (e.g. V1 neurons), and the likely explanation is that it occurs due to spike generation happening down the axon, at some electronic distance from the soma. We have now commented on this and referenced recordings with similar spike properties (subsection “V2b cell physiology does not distinguish between subtypes”).

9) Introduction, paragraph two: the statement that premotor neurons only belong to five superclasses is not 100% correct. Of the 10 subfamily, at least Di3 have been shown to contact motoneurons (Bui et al., 2013).

We appreciate your comment regarding this oversight, the text in paragraph two has been modified and the citation included.

10) Consider moving the 'in situ hybridization' section, table and figure to the Materials and methods or supplementary information. It is a necessary control, but interrupts the flow of information while reading the paper

Agreed; now the transgenic line validation text and figure have been moved to the supplementary information and is referenced in the first paragraph of the Discussion.

11) Results, second paragraph, and Figure 1: I was wondering whether illustrating the position of the soma of V2b interneurons with a coronal view would be more readable. Indeed, I found the use of false color depth-coded image in panel B difficult to read. Is the red background corresponding to the other half of the spinal cord? Because the color scale states L to R (supposedly left to right), while in the text it is written from medial to lateral. Showing a projection for a full segment for example on a coronal slice would help the reader.

This is a helpful suggestion and we have removed the false color image and included a z-projection image of a single hemisegment in Figure 1 instead. This looks much cleaner and allows for straightforward labeling of the CSF-cN and ISN neurons. The soma mediolateral position is not important to any points we make in the paper.

12) Subsection “Neurotransmitter expression defines subpopulations of V2b neurons”: would it be possible to provide data showing the expression of both GABA and glycine in the same neuron? Are purely GABAergic neurons existing? What is the functional consequences of having a co-release of GABA and glycine?

Thank you for this question. Unfortunately, the genetically encoded fluorescent reporters available to us limit our ability to show three color in-vivo images of Gad1b, Glyt2, and Gata3, and thus we have relied on cell counts and distributions to estimate neurotransmitter expression. However, the reviewer is completely justified in wondering whether V2b-GABA neurons exist. To the best of our ability we can only state the possibility that V2b-GABA exist, and if so at substantially smaller quantities than the V2b-gly and V2b-mixed populations. We have included a comment about this in the text in paragraph two of subsection “Neurotransmitter expression defines subpopulations of V2b neurons”.

13) Figure 7: the panel E is not easy to read and as such does not provide any obvious info. Is this panel really necessary?

We have removed this panel for clarity.

14) Please correct the opening sentence in subsection “V2b axons extend throughout the spinal cord” (probably a mix between two version of the same sentence).

Thank you for catching this typo, we have corrected it.

15) Paragraph three subsection “Neurotransmitter expression defines subpopulations of V2b neurons”: please correct sunclasses.

This has been corrected.

16) Subsection “Axonal morphology varies by subpopulation identity”: please define VeLD (this is done in the Discussion but not here).

A definition has been added at the first mention of VeLDs.

17) Figure 6—figure supplement 2: panel A the time scale on top might be wrong as the stim duration is 20ms while the scale indicates 0.4 sec?

Thank you for catching this mistake; we have changed the figure accordingly.